# Modelling the small-scale deposition of snow onto structured Arctic sea ice during a MOSAiC storm using snowBedFoam 1.0.

Océane Hames [1,2,*], Mahdi Jafari [2,*], David Nicholas Wagner [1,2], Ian Raphael [3], David Clemens-Sewall [3], Chris Polashenski [3,4], Matthew D. Shupe [5,6], Martin Schneebeli [1], and Michael Lehning [1,2]

[1]WSL Institute for Snow and Avalanche Research SLF, Davos, Switzerland
[2]CRYOS, School of Architecture, Civil and Environmental Engineering, EPFL, Lausanne, Switzerland
[3]Thayer School of Engineering, Dartmouth College, Hanover, New Hampshire, USA
[4]USACE-CRREL Alaska Projects Office, Fairbanks, Alaska, USA
[5]NOAA Physical Science Laboratory, Boulder, Colorado, USA
[6]Cooperative Institute for the Research in Environmental Sciences, University of Colorado Boulder, Boulder, Colorado, USA
[*]These authors contributed equally to this work.

**Correspondence:** Océane Hames (hames.oceane@gmail.com)

**Abstract.** The remoteness and extreme conditions of the Arctic make it a very difficult environment to investigate. In these polar regions covered by sea ice, the wind is relatively strong due to the absence of obstructions and redistributes a large part of the deposited snow mass, which complicates estimates for precipitation hardly distinguishable from blowing or drifting snow. Moreover, the snow mass balance in the sea ice system is still poorly understood, notably due to the complex structure of its surface. Quantitatively assessing the snow distribution on sea ice and its connection to the sea ice surface features is an important step to remove the snow mass balance uncertainties (i.e. snow transport contribution) in the Arctic environment. In this work we introduce *snowBedFoam 1.0.*, a physics-based snow transport model implemented in the open source fluid dynamics software OpenFOAM. We combine the numerical simulations with terrestrial laser scan observations of surface dynamics to simulate snow deposition in a MOSAiC (Multidisciplinary Drifting Observatory for the Study of Arctic Climate) sea ice domain with a complicated structure typical for pressure ridges. The results demonstrate that a large fraction of snow accumulates in their vicinity, which compares favorably against scanner measurements. However, the approximations imposed by the numerical framework together with potential measurement errors (precipitation) give rise to quantitative inaccuracies, which should be addressed in future work. The modelling of snow distribution on sea ice should help to better constrain precipitation estimates and more generally assess and predict snow and ice dynamics in the Arctic.

## 1 Introduction

Sea ice figures prominently in a broad range of environmental, socioeconomic and geopolitical applications (Huntington et al., 2017; Yumashev et al., 2017; mat, 2019). In the past years, the interest in sea ice has grown substantially as it seems to be strongly impacted by climate change: from the late 1970s until the present, the monthly average of the Arctic sea ice extent has undergone a significant reduction, with a downward trend of 13.1% per decade relative to the 1981-2010 average (Perovich

et al., 2020). Being able to predict the future evolution of this environment implies having a sufficient knowledge of the mechanisms underlying the ice generation and destruction dynamics.

Recently, different authors have stressed the need to have a robust quantification of the snow distribution since it seems to play a significant role in both the melt and growth of sea ice (Leonard and Maksym, 2011; Webster et al., 2018). The insulating capacities of snow as well as its high albedo can profoundly impact the internal energy budget of sea ice (Trujillo et al., 2016). Snow furthermore modifies the topography and the aerodynamic roughness of the sea ice surface, which impacts the turbulent energy transfer with the atmosphere (Andreas and Claffey, 1995). Lastly, snow can contribute to the ice mass as its weight reduces the freeboard, which may result in snow-ice formation (Ackley et al., 1990; Sturm and Massom, 2017); the importance of this process is expected to grow in the future with the thinning of the Arctic ice (Maslanik et al., 2007; Provost et al., 2017) and increasing precipitation (Bintanja et al., 2020). Hence, snow is presented as an integral component of the ocean - sea ice - snow - atmosphere system and quantifying its distribution is a key to a better understanding of the sea ice mass balance (Wever et al., 2020).

Also, the strong winds encountered in the Arctic environment lead to large uncertainties in both model projections and measurements of precipitation (Goodison et al., 1998; Wong, 2012; Boisvert et al., 2018), partly due to blowing snow being falsely detected as precipitation by snowfall sensors (Sugiura et al., 2003). Arctic precipitation estimates could be significantly improved by an accurate assessment of the snow transport and redistribution on sea ice, assuming that the other snow mass sink terms (e.g. sublimation/condensation and runoff) are known or negligible. Outside the melting season (polar night and adjacent months), the erosion of snow has been identified as the largest sink term and may cause up to a 50 % decrease of the total precipitated snow mass on sea ice (Leonard and Maksym, 2011). In comparison, processes such as sublimation, melt or condensation have shown to play small roles in the snow mass budget for the same period (Webster et al., 2021). Quantifying snow redistribution and connecting the snow mass balance to snowfall is a way to better constrain the current precipitation estimates and improve meteorological models.

Despite its importance, the current knowledge on sea ice snow depth distribution and its spatio-temporal evolution is limited due to sparse observational evidence (Trujillo et al., 2016; Sturm and Massom, 2017; Liston et al., 2018). The amount of snow that is accumulated, redistributed, sublimated and transported to the open water is determined by the complex interaction between snowfall, wind, and the presence and spacing of open leads; the contribution of each of these processes to the Arctic snow mass balance, however, is not yet fully resolved (Déry and Tremblay, 2004; Leonard and Maksym, 2011; Chung et al., 2011; Webster et al., 2021). In particular, surface features such as pressure ridges have a substantial effect on the snow distribution: by serving as aerodynamic obstacles, they enhance the deposition of drifting and blowing snow, which leads to the formation of depositional features such as drift aprons (Trujillo et al., 2016; Liston et al., 2018; Sommer et al., 2018). However, detailed quantitative spatial representations of the effect of such topographical structures are limited.

Several authors have investigated the snow distribution on sea ice in the recent years. Spatial measurements over a small-scale area of Antarctic sea ice (∼100×100 m) with terrestrial laser scanning by Trujillo et al. (2016) were used to characterize the influence of a storm on snow distribution patterns and their relation to the surface topography. Snow drifts were found to

form mostly behind the topographical obstacles, elongated along the dominant wind direction. To our knowledge, such spatial observations of snow deposition are non-existent in the literature for Arctic sea ice.

From a numerical perspective, Liston et al. (2018) recently applied a snow-evolution modelling system (SnowModel, Liston and Elder (2006)) to simulate snowdrifts and snow-depth distributions around sea ice pressure ridges. The authors ran a one-year simulation over an Arctic sea ice domain (1.5km $\times$ 1.5km) containing a ridged topography, which they tested against measurements. Results showed strong snow deposition behind the pressure ridges and snow-free sea ice at their top, with partial accumulation along the upwind side of the ridges. The employed topographical data was based on radar-derived images of ice as more precise spatial observations were not available. The study suggested that improvement could be made through the use of surface based and airborne light detection and ranging (lidar), which we achieve in the present work.

Generally speaking, the wind-induced snow transport processes near the ground (saltation) and at higher elevations such as suspension and preferential deposition (Lehning et al., 2008) are dominant drivers for the spatial variability of snow distribution at small scale (few meters to hundreds of meters) and shape the snow deposition patterns across various environments (Mott et al., 2018). Multiple studies describe and try to reproduce those small-scale snow deposition patterns at high resolution through the modelling of snow transport processes with detailed terrain-flow-particles interactions (e.g., Gauer, 2001; Mott and Lehning, 2010; Groot Zwaaftink et al., 2014; Wang and Huang, 2017).

In particular, the modeling framework developed by Comola et al. (2019) combines large eddy simulation (LES) for the flow and a Lagrangian stochastic model (LSM) for snow particle trajectories. Their model simulations demonstrate that different deposition patterns can emerge from different combinations of scale- and velocity-dependent dimensionless parameters. Thus, various factors can influence the spatial variability of snow distribution on sea ice and numerical modeling can help with understanding the dominant processes.

Herein we present *snowBedFoam 1.0.*, a new Eulerian-Lagrangian (E-L) snow transport solver implemented in the computational fluid dynamics (CFD) software OpenFOAM (or Open-source Field Operation And Manipulation, Weller et al. (1998)) that we employ to simulate the snow distribution patterns in a numerical domain containing a second-year sea ice topography with typical pressure ridges. Several data sets from the recent MOSAiC (Multidisciplinary Drifting Observatory for the Study of Arctic Climate) expedition (Shupe et al., 2020) are used for this purpose. The first data set consists of terrestrial laser scans (TLS) that we employ as a topographical base for the numerical domain in the simulations. We secondly use meteorological measurements from MOSAiC to set-up the wind and precipitation settings in the model and simulate specific atmospheric events. Finally, we compare the simulation output to real snow distribution measurements obtained by differencing successive digital elevation models (DEMs) of the snow surface. The TLS technology permits the survey of snow depth variability in a very high resolution (Prokop et al., 2008) and is particularly well suited for studying snow transport processes at small scales.

The novelties of the numerical approach developed in this study are multiple. First, to our knowledge, the OpenFOAM software has never been employed for the modelling of aeolian snow transport with such a detailed representation of the snow particle-bed interactions. Second, we initiate the use of the physical model of snow transport based on CFD and LSM (Groot Zwaaftink et al., 2014; Sharma et al., 2018; Comola et al., 2019) for sea ice applications, in addition to the integration of real snowfall and wind data as forcing parameters. Finally, the use of TLS elevation data as a base for the sea ice surface in

the simulation domain has not been achieved yet in the literature (Liston et al., 2018). Our aim in the present work is to assess
whether the snowBedFoam model is able to reproduce the small-scale snow distribution patterns found on Arctic sea ice, in
any qualitative or quantitative way. Only pure mechanical fluid-particle interactions are considered here, thus we distinctively
evaluate the impact of the horizontal snow transport on the sea ice snow mass balance at a given location. Thermal processes
such as the sublimation of blowing snow and snow at the surface, although having a big role in the snow mass budget at
certain spatio-temporal scales, are assumed to be negligible given the time period and location of interest (Chung et al. (2011);
Webster et al. (2021)). Previous studies demonstrated that the small-scale snow transport processes mainly drive the spatial
structure of snow distribution (Gerber et al., 2018), and it can be expected that a strict Eulerian-Lagrangian snow transport
model can reproduce the snow distribution patterns on sea ice qualitatively. An accurate quantitative evaluation of the snow
mass distribution is less likely, however, given the measurement uncertainties and modelling simplifications implied by our
numerical framework (see Section 2.4 Modelling assumptions for more details). This work is a first step towards the accurate
modelling of snow distribution on sea ice and could contribute in a larger frame to the improvement of precipitation and sea
ice mass balance estimates in model projections.

This article begins with the presentation of the MOSAiC measurements that were employed to produce the results in the
subsequent parts. In a second section, we describe the OpenFOAM snow transport model and the associated equations that
were implemented to reproduce the snow erosion and deposition processes on sea ice. Then, we present the details related to
the simulations, including the modelling assumptions and the numerical settings such as the forcing parameters. In the last
section, the results from the simulations are analyzed and compared to real DEM measurements. Discussion and concluding
remarks follow.

## 2   Data and Methods

Several processing steps were required to generate the results presented in this work. In this section we successively describe:
(1) the MOSAiC measurement campaign conducted on Arctic sea ice and the related data sets used here; (2) the implementation
of the snow transport model in the OpenFOAM software; (3) the assumptions and general settings of the simulations aiming to
reproduce the snow distribution on sea ice.

### 2.1   MOSAiC Campaign

Detailed observations of snow surface topography were conducted during the MOSAiC expedition, which took place in the
Central Arctic from September 2019 to October 2020 (Shupe et al., 2020). Trapped in the Arctic ice for a nearly full annual
cycle, the research vessel Polarstern (Knust, 2017) operated by the Alfred-Wegener-Institut Helmholtz-Zentrum für Polar-
und Meeresforschung (AWI) served as the center for data collection on a drifting sea ice floe: a kilometer-wide network of
monitoring stations was set up in its surroundings, allowing various and extensive measurements. In particular, observations
of the snow and sea ice properties and their governing processes were conducted year-round during the MOSAiC campaign
(Nicolaus et al., 2022).

This study focuses on repeated terrestrial laser scans of snow on second-year sea ice that were successively measured on November 6 and November 13, 2019 (Clemens-Sewall, 2021). The TLS-scanned region after processing covers a total area of 390 m x 340 m and its location relative to the MOSAiC sea ice floe is shown in Figure 1 (black frame). The topographical image in the background of Figure 1 is derived from aerial laser scan measurements taken on November 12, 2019 during an helicopter flight. It is only used to illustrate the relative sea ice floe location of the measurements used in this study. Throughout this article the term "scans" only refers to the TLS observations. A Riegl VZ-1000 scanner (RIEGL, 2017) was positioned at several locations with sufficient scan overlap to generate a three-dimensional (3D) cloud of points over the zone of interest. The emitter of the scanner was placed as high as possible (approx. 2.7 m above level surface) to reduce shadowing and the scan positions were recorded within an intrinsic project coordinate system. General details about the use of TLS for sea ice measurements can be found in Polashenski et al. (2012). Before its use, the raw point cloud was post-processed with the RiSCAN PRO v2.10. (RIEGL, 2020) software. Several corrections were made which included filtering, removal of outliers / non-static objects and the shift and rotation of tie points. The processed point clouds were then spatially interpolated into digital snow surfaces using the Triangulated Irregular Network (TIN) (or Delaunay) interpolation method, available in the QGIS open-source software (QGIS.org, 2022). Finally, the interpolated data was exported at grid scales between 20 cm to 1 m with QGIS. We chose to use the highest resolution to fully capture the small-scale structure of the surface. The two final DEMs reveal the surface position of the snow at two distinct times and their difference yields snow accumulation patterns that could be compared to the numerical results generated with snowBedFoam 1.0. Changes in snow-depth values were converted to comparative units of areal mass by multiplication with a constant snow density value of $210\,\mathrm{kg.m^{-3}}$ measured simultaneously during the campaign. This density value is derived from on-ground bulk measurements conducted in situ with an ETH tube and a snow density cutter (Haberkorn, 2019). This approach fails to take into account the spatial variability of snow density that is expected over the sea ice floe due to the combined effects of wind-induced snow redistribution and compaction, especially in the vicinity of topographical features affecting the wind flow-field (Leonard and Maksym, 2011; Sturm and Massom, 2017). MOSAiC research into this topic is currently ongoing and we expect some influence on the quantitative results presented in this paper.

Besides the sea ice surface topography, wind data was also required to accurately represent the mean flow field conditions in the numerical simulations. Wind information was collected using a meteorological station that was permanently installed on the investigated ice floe as part of the MOSAiC measuring network (Met City, green dot mark in Figure 1). An overview of the available meteorological observations is given in Shupe et al. (2022), and the data set with the raw near-surface meteorological flux tower measurements is publicly available (Cox et al., 2021). The meteorological station included three 3D Metek uSonic-3 Cage MP anemometers and three Vaisala HMT temperature and humidity sensors located at nominal 2 m, 6 m and 10 m heights. In particular, hourly data for wind direction and friction velocity were used to set up the flow parameters in our snow transport model. The MOSAiC atmosphere surface energy flux team performed the post-processing of the measurements, which included sonic data treatment (friction velocity), quality control processes, calibrations and filters. Corrective rotations of the winds to be relative to true north were also performed. A data journal article outlining all of the finer details of the Met City data set (including quality control procedures) should soon be published.

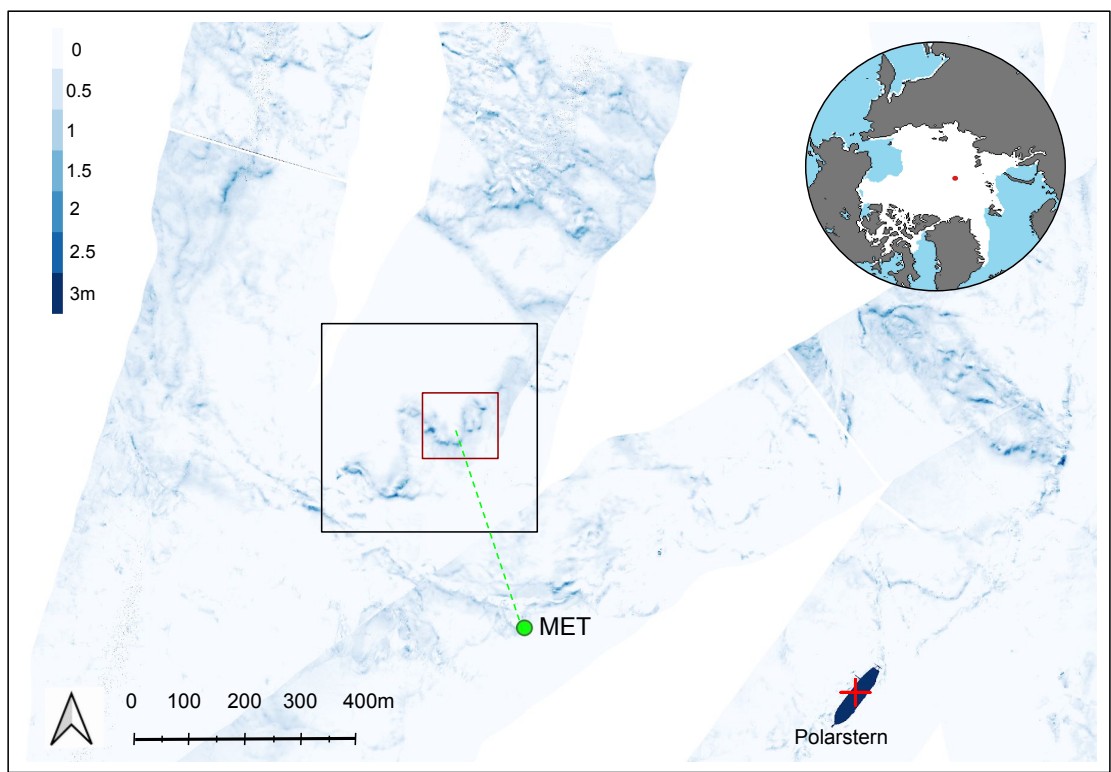

**Figure 1.** Aerial laser scan (ALS) of the sea ice floe taken on November 12, 2019 during a helicopter flight. The black rectangle pinpoints the zone covered by the terrestrial laser scans (TLS) which were used in the simulations. More particularly, the sea ice area framed within the red rectangle served as a base for the snowBedFoam 1.0. computational domain. The green dot mark highlights the position of the MOSAiC meteorological station where wind measurements were taken. The red cross points out the position of the Polarstern ice-breaker (Knust, 2017). The map on the top right corner shows the position of Polarstern on the day of the ALS measurement.

At last, we used precipitation estimates recorded during MOSAiC using ship-based Ka-Band ARM (Atmospheric Radiation Measurement) Zenith Radar (KAZR) reflectivities (Lindenmaier et al., 2019) at a 280 m range gate along with a reflectivity-based retrieval (Matrosov, 2007). KAZR devices provide several intervals of range (or time delay from transmission) in the vertical, within which returning radar signals are measured. Gating is used to isolate the echoes from different regions of distributed targets (Widener et al., 2012). Our steps proceed in the same way as Wagner et al. (2022), whose analysis suggests that the 280 m-ranged KAZR data measured during MOSAiC is on the whole reliable. More details concerning the choice of the radar range gate and the derivation of snowfall from observations can be found in their study. The radar recorded a 7-hour long snowfall event during the inter-TLS period (on November 11, 2019), which was included in OpenFOAM by releasing particles with log-normally distributed sizes in the domain at the average measured rate.

## 2.2 snowBedFoam 1.0. : an OpenFOAM snow transport model

### 2.2.1 Eulerian-Lagrangian solver

This section describes the three-dimensional Eulerian-Lagrangian model we developed in the open source platform Open-FOAM (The OpenFOAM Foundation, 2022) to simulate snow transport. OpenFOAM is a C++ object-oriented toolbox used to develop numerical solvers providing a solution to continuum mechanics problems (Weller et al., 1998), which is based on the finite volume method (FVM) for fluid dynamics computations. A comprehensive review of this field of research is beyond the scope of this paper. Further details on the implementation of the FVM in the software and on the various techniques developed over the years can be found in works such as Moukalled et al. (2015).

The so-called *DPMFoam* solver originally implemented in OpenFOAM version 2.3.0. was adapted to simulate the aeolian transport of snow particles. This multiphase flow solver handles coupled Eulerian-Lagrangian phases, which involves a finite number of particles spread in a continuous phase (OpenFOAM API Guide, 2006a). It is based on the Lagrangian particle tracking (LPT) technique called discrete particle method (DPM), which models the system at the micro-mechanical level and tracks the motions of all the particles, or agglomerates of particles (parcels) (Richards et al., 2004). LPT modelling is the most straightforward and effective approach to obtain the deposition and erosion locations of snow (Wang and Huang, 2017). In DPMFoam, the Eulerian continuum equations including particle volume fraction are solved for the fluid phase whereas Newton's equations for motion are solved to determine the trajectories of the particles (parcels). Fernandes et al. (2018) tested the solver against various data types and found a good agreement with results from the literature. The governing equations for particle and fluid-phase systems employed in DPMFoam are described hereafter.

*Governing equations for the particle system*

A particle in movement may exchange momentum and energy with other particles, domain surfaces (internal or external) or with the surrounding fluid (Fernandes et al., 2018). Most of the typical forces acting on a particle in a granular flow can be included within snowBedFoam 1.0., if required. We chose to adopt a two-way coupling method for our sea ice simulations considering reciprocal action between fluid and particles, while other interactions (e.g. particle - particle) were neglected. In this study, the snow particles were assumed to be subject to gravity and atmospheric drag, and the governing equation for particle motion can be expressed through Newton's second law :

$$m_i \frac{d\boldsymbol{U}_i^p}{dt} = m_i \boldsymbol{g} + \boldsymbol{F}_d \tag{1}$$

where $\boldsymbol{U}_i^p$ is the particle velocity, $\boldsymbol{F}_d$ is the drag force, $\boldsymbol{g}$ the gravity acceleration vector and $m_i$ is the mass of a spherical particle. The latter is formulated as $m_i = \frac{1}{6}\pi\rho_p d_i^3$, with $d_i$ the diameter and $\rho_p$ the density of the particle. The drag force stems from the particle–fluid interaction and is proportional to the relative velocity between the phases. Numerous drag models are available in the OpenFOAM framework: we chose to adopt the commonly used sphere-drag model based on the assumption of solid spheres (OpenFOAM API Guide, 2006c). If $\boldsymbol{U}^f$ represents the fluid velocity, then the corresponding drag force exerted

on a spherical particle is defined as

$$\boldsymbol{F}_d = \frac{3m_i C_D Re_p \mu}{4\rho_p d_i^2}(\boldsymbol{U}^f - \boldsymbol{U}_i^p) \tag{2}$$

where $\mu$ represents the dynamic viscosity and $Re_p$ is the particle Reynolds number

$$Re_p = \frac{d_i|\boldsymbol{U}^f - \boldsymbol{U}_i^p|}{\nu} \tag{3}$$

characterizing the relative importance between the inertial and viscous forces acting on a particle, with $\nu$ the kinematic viscosity. $C_D$ is the so-called drag coefficient and is defined as follows

$$C_D = \begin{cases} 0.424 & \text{for } Re_p > 1000 \\ \frac{24}{Re_p}\left(1 + \frac{1}{6}Re_p^{2/3}\right) & \text{for } Re_p \leq 1000 \end{cases} \tag{4}$$

Information about the forces and coupling modes available for particle modelling in OpenFOAM can be found in OpenCFD Ltd (2018) and OpenFOAM API Guide (2006b).

### Governing equations for the fluid-phase system

The flow equations implemented in DPMFoam involve the fluid-phase volume fraction $\epsilon_f$ in an Eulerian cell expressed as:

$$\epsilon_f = \max\left(1 - \frac{1}{V_{cell}}\sum_{i=1}^{N_p} f_i^p V_i \ , \ \epsilon_{fmin}\right) \tag{5}$$

with $V_{cell}$ the cell volume, $f_i^p$ the fractional volume of a particle $i$ located in the cell under consideration, $V_i$ the volume of particle $i$ and $\epsilon_{fmin}$ a very small value which limits the cell from being fully occupied by a particle. $N_p$ represents the total number of particles present in the computational cell. The Navier-Stokes equations (Equation 6) and the volume-averaged continuity equation (Equation 7) are solved for an incompressible fluid phase in the presence of a secondary particulate phase:

$$\frac{\partial(\epsilon_f \boldsymbol{U}^f)}{\partial t} + \nabla \cdot (\epsilon_f \boldsymbol{U}^f \boldsymbol{U}^f) = \mathscr{P} - \nabla P + \nabla \cdot (\epsilon_f \boldsymbol{\tau}_f) - \boldsymbol{F}_p + \epsilon_f \mathbf{g}, \tag{6}$$

$$\frac{\partial \epsilon_f}{\partial t} + \nabla \cdot (\epsilon_f \boldsymbol{U}^f) = 0 \tag{7}$$

where $P$ is the modified pressure ($p/\rho_f$, with $\rho_f$ being the fluid density) and $\boldsymbol{\tau}_f$ is the fluid-phase viscous stress tensor. $\mathscr{P}$ represents the imposed large-scale driving force in the streamwise direction which was added by the authors within the core code of the solver and used as driver for the flow in the simulations presented hereafter. It is described as:

$$\mathscr{P} = -\frac{1}{\rho_f}\frac{\partial \widetilde{p_\infty}}{\partial x} = \frac{u_*^2}{L_z} \tag{8}$$

with $L_z$ the vertical extent of the numerical domain and $u_*$ the surface friction velocity. The sink term $\boldsymbol{F}_p$ in the momentum equation (Equation 6) accounts for the two-way coupling between the fluid-phase and the particles. As the fluid drag acting on each particle is known, this term is computed as a volumetric fluid–particle interaction force given by:

$$\boldsymbol{F}_p = \frac{\sum_{i=1}^{N_p} \boldsymbol{F}_{d,i}}{\rho_f V_{cell}} \tag{9}$$

where $\boldsymbol{F}_{d,i}$ is the drag force exerted on particle $i$. $\boldsymbol{F}_p$ is here presented in a discretized form.

### 2.2.2  Snow-wind interaction model

Similarly to sand transport (Bagnold, 1941), the aeolian transport of snow particles can be classified into three modes (e.g., Pomeroy and Gray, 1995; Aksamit and Pomeroy, 2016), namely: 1) *creeping*, which consists of the rolling of particles along the surface; 2) *saltation*, which occurs when particles follow ballistic trajectories and involves mechanisms such as aerodynamic lift along with rebound and entrainment (splash) of snow grains; 3) *suspension*, which entails the same mechanisms as saltation but applies to smaller grains transported at higher elevations and over larger distances. We developed several utilities within the OpenFOAM Lagrangian library to introduce the processes of aerodynamic lift, rebound and splash of particles. Thus, our new modelling tool simulates the redistribution of snow through saltation (drifting snow) and suspension (blowing snow).

The governing equations that were implemented in the solver stem from snow transport-related publications from various authors (Anderson and Haff, 1991; Groot Zwaaftink et al., 2014; Comola and Lehning, 2017). We employed a similar set of equations for particle-flow interaction to the one implemented in the large eddy simulation - Lagrangian stochastic model (LES-LSM) which was used to generate publications such as the ones of Groot Zwaaftink et al. (2014), Comola et al. (2019) or Sharma et al. (2018). The equations relevant for our snow transport model are summarized subsequently.

*Aerodynamic entrainment*

The early work of Bagnold (1941) on sand transport instituted the concept of aerodynamic entrainment (or lift) of particles. It occurs when the wind flow has sufficient momentum to lift up particles from the surface, namely when the fluid surface shear stress $\tau_{f,surf}$ exceeds a certain threshold value $\tau_{th}$. Findings on wind-driven sand transport stay relevant to snow and constituted the groundwork for many snow-wind interaction studies (e.g., Schmidt, 1986; Pomeroy and Gray, 1990; Li and Pomeroy, 1997). Experimental results on snow transport initiation by Clifton et al. (2006) showed a good agreement with Bagnold's initial formulation of shear stress threshold, using $A = 0.18$ as an empirical constant:

$$\tau_{th} = A^2 g \langle d_p \rangle (\rho_p - \rho_f) \tag{10}$$

where $\langle d_p \rangle$ is the mean particle diameter. In each grid cell, the number of particles aerodynamically entrained by the fluid at each timestep, $N_{ae}$, linearly increases with the excess shear stress according to the formulation of Anderson and Haff (1991):

$$N_{ae} = \frac{C_e}{8\pi \langle d_p \rangle^2} (\tau_{f,surf} - \tau_{th}) \Delta x \Delta y \Delta t \tag{11}$$

where $C_e$ is an empirical parameter set to 1.5 (Groot Zwaaftink et al., 2014), $\Delta x$ and $\Delta y$ are the grid dimensions in the streamwise/spanwise directions and $\Delta t$ is the simulation timestep. Once $N_{ae}$ is determined, properties such as the particle

diameter, initial velocity magnitude and ejection angle are sampled from statistical distributions according to Clifton and Lehning (2008). More details can be found in their work.

### Rebound and splash entrainment

Depending on its path, a snow particle present in the fluid might hit the surface upon which it can not only rebound -defined as *rebound* entrainment- but also eject other particles from the bed to the overlying fluid, defined as *splash* entrainment. The

probability $P_r$ that the snow particle rebounds when impacting the bed is given by Anderson and Haff (1991) as follows

$$P_r = P_m(1 - e^{-\gamma v_i}) \tag{12}$$

where $P_m$ is the maximum probability equal to 0.9 for snow (Groot Zwaaftink et al., 2013), $\gamma$ is an empirical constant equal to 2, and $v_i$ is the velocity magnitude of the impacting particle. When rebounding, the particle is assumed to have a velocity magnitude of $v_r = 0.5v_i$ (Doorschot and Lehning, 2002) and the rebound angle is determined from a statistical distribution

according to Kok and Renno (2009).

Concerning the splash entrainment, the number of particles ejected from the bed $N_{splash}$ is defined as the minimum between $N_E$ and $N_M$ whose expressions are (Comola and Lehning, 2017):

$$N_E = \frac{(1 - P_r\epsilon_r - \epsilon_{fr})d_i^3 v_i^2}{2\langle v\rangle^2(\langle d\rangle + \frac{\sigma_d^2}{\langle d\rangle})^3 \left(1 + r_E\sqrt{5[1 + (\frac{\sigma_d}{\langle d\rangle})^2]^9 - 5}\right) + 2\frac{\phi}{\rho_p}} \tag{13}$$

$$N_M = \frac{(1 - P_r\mu_r - \mu_{fr})d_i^3 v_i cos\alpha_i}{\langle v\rangle^2(\langle d\rangle + \frac{\sigma_d^2}{\langle d\rangle})^3 \left(\langle cos\alpha\rangle\langle cos\beta\rangle r_M\sqrt{[1 + (\frac{\sigma_d}{\langle d\rangle})^2]^9 - 1}\right)} \tag{14}$$

$N_M$ and $N_E$ are the number of ejections predicted by the momentum and energy balance, respectively. In Equation 13, $\epsilon_{fr}$ and $\epsilon_r$ are the fractions of impact energy lost to the bed and kept by the rebounding particle, respectively. $\mu_{fr}$ and $\mu_r$ are their equivalent for momentum in Equation 14. $\langle d\rangle$ and $\sigma_d$ are the mean and standard deviation of ejecta's diameter, $\langle v\rangle$ its mean velocity and $\alpha$ and $\beta$ the horizontal and vertical ejection angles. $\phi$ is the cohesive bond exerted on a particle by its

neighboring particles. $r_M$ and $r_E$ are correlation coefficients linking mass and velocity. More details about the derivation of these formulations can be found in the work of Comola and Lehning (2017). Similarly to the aerodynamic entrainment, the characteristics of the splashed particles are randomly sampled from statistical distributions. Overall, details about the equations of the snow surface-flow interaction can be found in the Supplementary Material of the study by Sharma et al. (2018).

## 2.3   Simulation Settings

### 2.3.1   Numerical Domain

The mesh employed for our OpenFOAM simulations on sea ice was generated based on the first set of TLS scans dating from November 6, 2019. The red-framed area in Figure 1 pinpoints the part of the ice surface scanned with ground-based lidar that was used as a base for the numerical domain ($\sim$110 m$\times$120 m). This zone was selected due to its central position in the DEM along with the variation in height, shape and orientation of its ridged ice. A 10 m wide buffer zone was created between the area of interest and the lateral boundaries to limit the influence of the latter on the simulated flow. The vertical extent of the domain was set to 15 m, with a maximum pressure ridge height of 3.1 m. We chose to limit the dimensions of the numerical domain to perform less computationally intensive simulations. However, the snowBedFoam 1.0. solver could be run on larger-scale areas if required.

The numerical domain with gridded sea ice-topography was generated with input data in the STereo Lithography (STL) format. STL files describe the surface geometry of a three-dimensional object without any representation of other attributes (e.g. color, texture). Linear grid stretching was applied in the vertical direction to insure sufficient grid points in the saltation region while limiting the computational costs in the upper part of the domain; the vertical grid spacing $dz$ ranges between $dz = 0.5$ m for the coarsest grid resolution and $dz = 0.1$ m for the finest grid resolution near the surface. In the streamwise and cross-stream directions, a uniform grid was applied with an average resolution $dx$ , $dy = (0.5$ m$)$. The cell size was made fine enough to accurately capture the topographical features found at the sea ice surface, while keeping a reasonable mesh size to perform the computations. The final mesh is composed of 2 million hexahedral cells with an average size of 0.125 m$^3$, a minimum cell size value of 0.019 m$^3$ and a maximum cell size of 0.277 m$^3$.

### 2.3.2   Boundary conditions

The set of simulations in this work were ran by imposing periodic boundary conditions (PBCs) at the lateral sides of the domain (Fig. 2). PBCs are based on the inter-connection of the mesh elements on opposite faces: OpenFOAM treats the flow at a periodic boundary as if the opposing periodic plane was a direct neighbor to the cells adjacent to the first periodic boundary. This has the advantage to keep a reasonable domain size while guaranteeing fully developed velocity profiles within the domain. For Lagrangian applications, PBCs imply that a particle reaching a lateral boundary is directly re-injected in the domain through the corresponding cell of the connected periodic plane. In other terms, this approach translates into snow being injected at the upwind domain boundary. Regarding the sea ice boundary at the bottom, no-slip and impermeability boundary conditions were imposed at its surface for the horizontal and vertical velocity components, respectively. Moreover, the region near the surface boundary was modelled through standard wall functions. At the top boundary the horizontal components of velocity were set to Neumann zero-gradient BCs and the vertical component was set to zero.

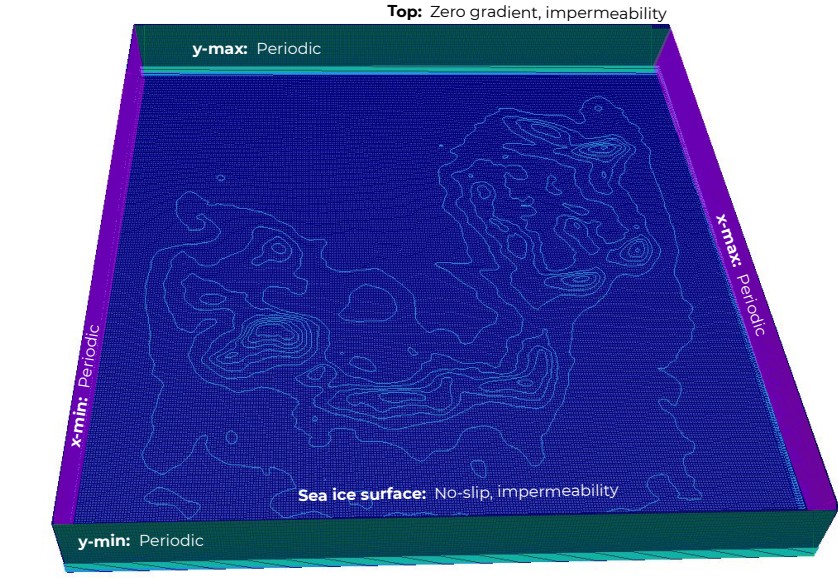

**Figure 2.** Boundary conditions applied at the six domain patches in the OpenFOAM snow transport simulations. Similar colours are used to represent the corresponding periodic patches.

### 2.3.3 Numerics

The Reynolds-Averaged Navier-Stokes (RANS) method and $k - \epsilon$ closure model were employed to solve the set of equations for a flow with neutral stratification (Zhang, 2009). We refer to the introductory paper by Launder and Spalding (1974) and to flow dynamics reference books (e.g. Stull (1988)) for additional details on these particular CFD techniques. To discretize the equations, the Gauss linear and bounded Gauss upwind schemes, respectively, were used for the terms with gradient and divergence operations while the Euler scheme was chosen for the discretization of the transient terms (Moukalled et al., 2015).

To assure a faster numerical convergence and minimize the computational effort, the model was initialized with a standard logarithmic wind profile within the boundary layer.

     For the flow time step, we make use of an automatic control called "*adjustableRunTime*" available in OpenFOAM, which adapts the time step based on a maximum Courant number value defined by the user (set to 1 in our case). The Courant number as the stability criterion is defined as the product of fluid velocity and time step divided by the numerical cell length scale.

Thus, the Eulerian time step changes with the grid size and local flow velocity. More information regarding the adjustable time step method for the flow is available in Moukalled et al. (2015) and Jafari et al. (2022). Within one Eulerian time step, there are several smaller Lagrangian time steps, which allow to adequately capture the parcel motion. The so-called "face-to-face tracking algorithm" (Peng, 2008; Macpherson et al., 2009) adapts the Lagrangian time step depending on the crossed boundaries and automatically limits its value to appropriately capture micro-scale processes such as the rebound-splash of

snow grains at the surface.

The Eulerian quantities were estimated at the particle location using the so-called "*cellPoint*" interpolation method in OpenFOAM, which performs linear interpolation with inverse distance weighing based on the closest cell point values (e.g., Moukalled et al., 2015; Leonard et al., 2021).

It should be mentioned that the particle mass balance was checked for each of the implemented submodels (including the injection model for precipitation). Moreover, a validation study against the well-established LES-LSM was conducted after the implementation of the new snowBedFoam 1.0. model (not shown here). The results for wind-field and particle mass flux were in good agreement despite fairly distinct grid structure and boundary conditions. This has further strengthened our confidence in the validity of the OpenFOAM model.

### 2.3.4 Particle and Flow Properties

The flow properties were set in the simulations to approximate the mean field conditions for the period between the two MOSAiC scans of interest. For this purpose, the meteorological time series were decomposed into four distinct periods corresponding to dominating phases of wind velocity for which the wind speed and direction were averaged (colored areas I-IV in Figure 3), following a successful strategy introduced by Lehning et al. (2008). This approach limits the amount of computations needed to be performed. These specific intervals were selected based on the friction velocity and duration: any time period with a minimum span of 3 hours and a friction velocity higher than 0.2 m.s$^{-1}$ was considered as having a substantial influence on the snow deposition patterns. The friction velocity threshold was chosen based on the publications from various authors who found that snow transport was initiated above this value (He and Ohara, 2017; Clifton et al., 2006; JDoorschot et al., 2004). The model was initialized with wind fields characteristic for the respective events to reproduce as accurately as possible the measured snow distribution. Regarding precipitation, the MOSAiC meteorological instruments recorded a 7-hour long snowfall on November 11 (Figure 3, event IV), which we reproduced in the model.

A total of four simulations mimicking the selected time periods were run with OpenFOAM, whose forcing parameters and time spans are summarized in Table 1. Symbols I to IV in the first column connect with the characters displayed at the top of Figure 3. In the last two columns the values for friction velocity and wind direction correspond to the averages over each event. In order to limit the computational effort, the total mass deposition values were extrapolated to the duration of the measured wind events using the particle deposition rates after 1000 s of simulation, which represents an approximation to a steady state situation (i.e., small variations of the surface friction velocity and total snow mass aloft in the domain, which indicate a flow-particle system at equilibrium).

In this context, we verified that the cell deposition rates and distribution patterns do not show strong trends for longer runs. Based on the variations of the total snow mass aloft in the domain, the time step at which a steady-state is reached is identified (stable saltation flux) and the deposition and erosion rates in each cell are derived based on the snow mass distribution results for the steady-state period until the end of the simulation. The estimated snow distribution at the end of a given wind period are further obtained by multiplying the rates with the total duration of the simulated event.

The snow properties in the simulations (Table 2) were selected on the basis of previous values reported in the literature. All particles were assumed to be spherically shaped and constituted of pure ice. From a dimensional point of view, the mean grain

**Table 1.** Snow transport and precipitation events identified during the period between the successive laser scans (2019).

| Event | Start Time | End Time | Friction Velocity | Wind Direction |
|:-:|:-:|:-:|:-:|:-:|
| – | dd.mm HH:MM | dd.mm HH:MM | m.s$^{-1}$ | ° |
| I | 06.11 12:00 | 07.11 00:00 | 0.24 | 85 |
| II | 08.11 16:00 | 09.11 13:00 | 0.27 | 321 |
| III | 10.11 23:00<br>11.11 18:00 | 11.11 11:00<br>13.11 07:00 | 0.30 | 183 |
| IV | 11.11 11:00 | 11.11 18:00 | 0.48 | 179 |

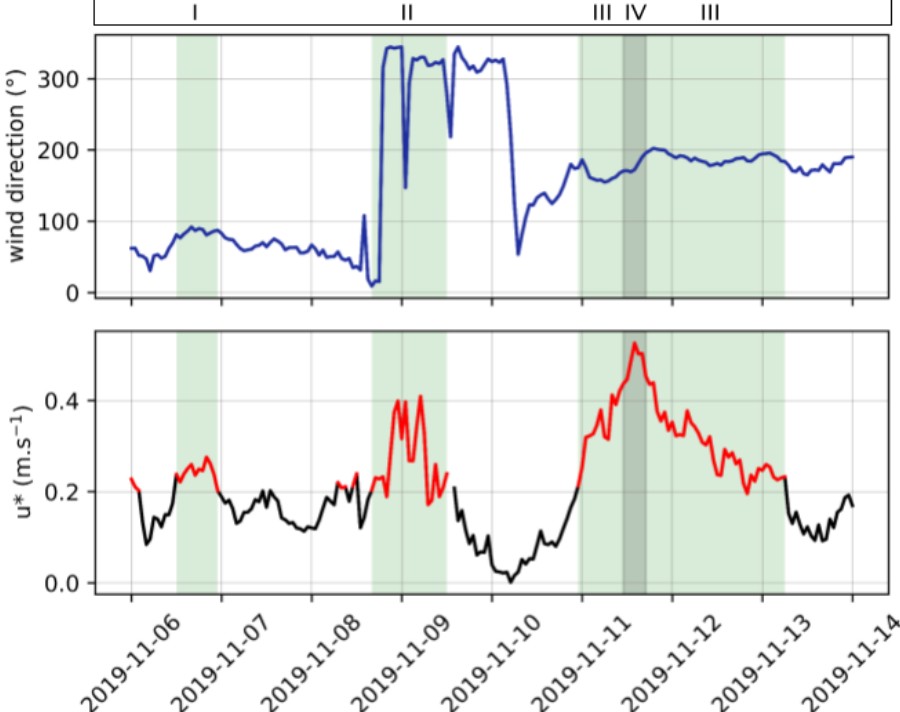

**Figure 3.** Time series for wind direction (top) and surface friction velocity at 2m (bottom) measured in the period between the successive TLS measurements. The values displayed are an average of measurements taken at 2 m, 6 m and 10 m height. The green-colored areas (I to III) correspond to the time periods for which snow movement was assumed to occur. The darker area (IV) shows the time span during which precipitation was measured by the radar. Source: Cox et al. (2021).

size is equivalent to the one defined in Comola and Lehning (2017) in their snow splash entrainment model; it is also in line with the findings of Nemoto and Nishimura (2004) and Gromke et al. (2014) who measured dimensions of transported snow grains. The particle size distribution of snowflakes is approximated using the log-normal law, although this parameter proved

**Table 2.** Particle properties employed in the sea ice OpenFOAM simulation.

| Variable | Symbol | Value | Unit |
|---|---|---|---|
| Mean diameter | $d_m$ | 0.2 | mm |
| Minimum diameter | $d_{min}$ | 0.05 | mm |
| Maximum diameter | $d_{max}$ | 0.5 | mm |
| Standard deviation of diameter | $d_{std}$ | 0.05 | mm |
| Particle density (ice) | $\rho_p$ | 900 | $kg.m^{-3}$ |
| Snowfall rate | I | 0.39 | $mm_{water}.h^{-1}$ |
| Shear stress threshold constant | A | 0.2 | - |
| Bed cohesion | $\phi$ | $10^{-9}$ | J |
| Time of saltation initiation | $t_{init}$ | 100 | s |

not to affect the deposition patterns (Wang and Huang, 2017). For simulation IV in particular, the average precipitation rate was set according to KAZR snowfall retrievals (280 m range gate) from the detected storm event (Wagner et al., 2022).

The last two parameters relate to the surface interaction equations of the model. *A* is a constant used in the shear stress threshold formulation (Equation 10), which we set equal to 0.2 for snow based on wind-tunnel experiments (Clifton et al., 2006). The bed cohesion parameter $\phi$ involved in the ejection entrainment (Comola and Lehning, 2017) represents the mean bounding energy between the grains of the snowbed and is usually found in the range $10^{-10}-10^{-8}$ J (Gauer, 2001). A sensitivity analysis showed that this parameter does not substantially influence the results for the range of friction velocities employed
in this work; therefore, an intermediate value ($10^{-9}$ J) in the above-mentioned range was selected. Details regarding the effect of the particle diameter involved in the shear stress threshold formulation (Equation 10) can be found in the work of Melo et al. (2022) who studied the impact of mean grain size on saltation fluxes using LES-LSM simulations. All the simulations were initiated without particles for the first 100 s to ensure a fully developed flow-field.

## 2.4    Modelling assumptions

To reduce the computational complexity while still enabling relevant modelling, several assumptions were made. First, only gravity and fluid-particle drag were considered in the force balance to solve the grain trajectories (Equation 1). As commonly done in snow transport applications (e.g. Gauer (2001)) we neglect the other small particle–fluid interaction forces commonly found in nature (e.g. buoyancy, pressure gradient) as well as the inter-particle collisional forces. Second, the mass loss due to snow sublimation, either from the surface or from blowing snow particles, is not taken into account in our model. The
flow-particle interactions are purely mechanical and do not encompass temperature and moisture feedbacks. Snow sublimation

may be of significant importance for the snow mass distribution depending on the observed spatial and temporal scales (Mott et al., 2018). However, given the location and time-scale of interest of this study, we assume the sublimation mass sink to be negligible (Chung et al., 2011; Webster et al., 2021; Wagner et al., 2022). Third, we adopted a parcel-based approach meaning that the particle population was represented by clouds of particles (parcels) with homogeneous properties. This technique does not take into account the clustering effect of particles represented by a single parcel (Radl and Sundaresan, 2013) and tends to minimize the variety in particle properties.

Turning now to Eulerian-phase modelling, the RANS method that we employed is based on ensemble-averaged governing equations and cannot predict the local unsteadiness (eddies) in the flow (e.g. Stull (1988)). This approach only approximates the temporal dynamics observed in turbulent flows and thus the representation of the intermittent snow transport is only partial, which can introduce bias in the modelled snow mass flux and ultimately in the distribution patterns.

Additional limitations arise from the forcing of the model. A one-week period with heterogeneous wind is represented by four discrete events only, based on a constant friction velocity threshold (Sect. 2.3.4). Although long for the evaluation of blowing snow, we defined our simulation period based on the terrestrial laser scans measurements available during the MOSAiC campaign. To be able to compare the model results to snow distribution data obtained in the field, we had no choice but to simulate a full week of snow redistribution on sea ice. The onset of snow transport is in reality time- and environment-dependent and the periods with saltation may have been poorly estimated. We are aware that these restrictive representations of natural phenomena may have limited the performance of the results presented hereafter. However, they remain acceptable for a semi-quantitative comparison such as the one performed in this work.

## 3 Results

### 3.1 Snow distribution patterns per event

Figures 4 and 5 report the results, respectively, of the extrapolated areal snow-mass distribution and surface friction velocity produced by our OpenFOAM snow model for the four selected atmospheric events (I−IV). The term "extrapolated" refers to the snow-mass distribution values obtained after multiplying the simulated snow transport rates by the total duration of each snow transport event. The wind direction is represented in each sub-figure by an arrow. For all the numerical simulations, the effect of the sea ice topography on the snow surface distribution was clearly captured; the snow particles appear to deposit on the lee-ward side of the pressure ridge whereas they get eroded on its wind-ward side and at the top. This spatial pattern agrees with previous observations suggesting that snow drift aprons mostly form in the lee of sea ice blocks and pressure ridges (Sturm et al., 2002; Massom et al., 2001). Also, it is in line with snow distribution measurements on an Antarctic sea ice floe which revealed strong deposition behind topographic obstacles according to the predominant wind direction (Trujillo et al., 2016).

Increased snow deposition is found in zones of highly-turbulent, decelerated flow, which typically appear behind the topographical obstacles; such wake regions are colored in blue in the surface friction velocity plots (Fig. 5). Alternatively, a fluid that is deviated by the terrain usually accelerates, which strengthens its forces on the ground and enhances snow erosion, as

**Table 3.** Numerical quantitative results for the OpenFOAM sea ice simulations per identified event (I-IV) between November 6 and 13, 2019. Relative areal proportion, average values and overall range for snow deposition and erosion are presented.

| Event | Snow deposition | | | Snow erosion | | |
|---|---|---|---|---|---|---|
| | mean $kg.m^{-2}$ | maximum $kg.m^{-2}$ | area % | mean $kg.m^{-2}$ | maximum $kg.m^{-2}$ | area % |
| I | 0.2 | 9.4 | 45.4 | -0.2 | -15.9 | 41.0 |
| II | 0.3 | 17.0 | 53.2 | -0.4 | -45.2 | 46.7 |
| III | 1.3 | 109.8 | 51.3 | -1.3 | -122.8 | 48.7 |
| IV | 3.5 | 126.4 | 90.3 | -4.2 | -56.5 | 9.7 |

observed over the ridge and along steep slopes. This connection between surface friction velocity and snow-mass distribution is highlighted through the comparison of their corresponding surface patterns. The surface friction velocity results in Figure 5 show elongated, low-velocity streaks for events I to IV. They are especially apparent in Figure 5-I due to the lower range of surface shear stress. Similar structures were identified in previous RANS-modelling studies (e.g., Hesp and Smyth, 2017; Ivanell et al., 2018; Wagenbrenner et al., 2019). A thorough analysis of our simulated wind-fields revealed the presence of counter-rotating vertices in the lee-side of our sea ice topography, whose convergence could be at the origin of such streaks (Hesp and Smyth, 2017). Our supposition is that these streaks are naturally induced by the sea ice ridge, but that they may be over-represented in our simulations due to the type of boundary conditions and discretization scheme employed. A second-order linear upwind discretization scheme was used for the divergence term in the momentum equation (convection), which has shown to produce broader and longer low-velocity streaks compared to other discretization schemes (Wagenbrenner et al., 2019). Moreover, periodic boundary conditions imply that an object reaching the downwind boundary of the domain is transferred to the upwind side at the next time step, thus reproducing a zone of low velocity at the windward side of the sea ice topography. This could resemble the real sea ice terrain, where the simulated ridge is located downwind of other ridges impacting the flow. Such numerical uncertainties can be hardly avoided and are difficult to quantify, as there is very little guidance in terms of the realistic representation of these streamwise flow features (Wagenbrenner et al., 2019).

The snow distribution results (Fig. 4) as well as the average (extreme) values for snow deposition and erosion (Table 3) are variable between the simulations: factors such as the slope angle encountered by the flow, the duration of the wind events and the magnitude of the fluid forcing can explain these differences. The mean snow deposition and erosion are specifically enhanced for the event with precipitation (IV). The precipitation particles combined with strong flow seem to favour the uplift of the snow grains when hitting the surface. Figure 4-IV shows that deposition is ubiquitous over the domain (93.4% of the area) but remains stronger at the lee of the ridge. Quantitatively speaking, the extent of the areal snow-mass range indicates that event I is the least influential on the snow distribution as it represents at most 50% of the redistribution values obtained in the other cases. This most probably stems from the lower duration and friction velocity characteristic of that period. Contrastingly, simulations III and IV have largest erosion and deposition values and therefore have the most influence on the combined snow distribution results.

## 3.2 Comparison to MOSAiC measurements

After evaluating the numerical results for each event individually, the left of Figure 6 displays the snow distribution patterns obtained by their combination; the right-side illustration shows the snow-mass distribution changes measured during MOSAiC over the period of interest (November 6−13).

A qualitative analysis reveals that a nearly uniform layer of deposited snow attributable to event IV has formed over the numerical domain, which contrasts with measurements where snow mass changes are close to zero in flat areas. This discrepancy put aside, the model results satisfyingly agree with the measurements on the location of enhanced snow deposition: circles **A** to **D** in Figure 6 highlight zones of accumulated snow that appear in both the model and measurements. Yet, there are features that were not reproduced by OpenFOAM: the northern side of the ridge (above **B**) and the zone between **C** and **D** show enhanced snow accumulation, which is almost absent in the combined simulation output. The closest similarity in deposition patterns (neglecting the homogeneously higher deposition) is found with simulation IV (Fig. 4), which shows snow accumulation between **C** and **D** and more elongated patches parallel to the predominant wind direction. Thus, a strong flow together with precipitation seems to mostly account for the measured distribution. Regarding erosion, the wind-ward side and top of the ridge show snow depletion in both simulated and measured data, although to different extents. The measured erosion is more accentuated, especially in between the fragments of the ridged ice (**B**-**C**, **C**-**D**) and in the southern area of the domain. We also find that some deposition patterns revealing the micro-relief of the ice surface were modelled in the flat areas around the ridge but are not as apparent in the measurements: they emerge only in localized sections (e.g., South-East of the domain). The relative proportion of erosion to the total area represents 13.0% and 62.7% in the model and measurements, respectively.

Quantitatively, our model appears only partially successful in predicting the snow mass changes over the numerical domain. Figure 7 shows the probability distribution of both measurements (blue) and simulation results (green). SnowBedFoam 1.0. underestimates the erosion found in the measurements and yields a higher proportion of cells with moderate snow deposition (between 0 and 10 $kg.m^{-2}$). In general, the measured snow mass change has more extreme values than in the model: it has a standard deviation of 10.7 $kg.m^{-2}$ compared to 7.25 $kg.m^{-2}$ for snowBedFoam. The mean snow mass change value is positive for the combined output (2.77 $kg.m^{-2}$), due to the addition of precipitation particles in simulation IV: the periodic set-up implies that any added mass is systematically re-injected through the lateral patches of the domain and ends up depositing on its surface. Note that this mean value approximately corresponds to the mass per surface computed using the average precipitation rate and the surface area of the domain (2.73 $kg.m^{-2}$). The measurements reveal an overall slight erosion (-0.6 $kg.m^{-2}$) : the snow has been transported outside of the domain. A number of reasons may have caused these dissimilarities, which will be discussed in the next section.

## 4  Discussion

Figure 6 reveals that the main zones of snow deposition and depletion were captured by snowBedFoam, even though it simulates a higher snow deposition on average in the flatter area of the domain. The deposition patterns scanned during MOSAiC show some satisfying agreement with the simulations, except for a few patches of enhanced accumulation, which are missing in

the numerical results. Regarding erosion, the model underestimates it in most locations, although spatial erosion patterns are qualitatively well captured at the ridge. The locations with lower erosion appearing in Figure 4 were likely damped by the precipitation particles settling on the surface (Event IV). The measurements more frequently observed larger snow mass changes compared to the simulations. There may be multiple reasons for these results.

1. *Simplification of wind transport*: the snow transport by the wind may be oversimplified due to a limited representation of the real aeolian conditions. For the selected events, the ratio of the standard deviation between the measurements to the average value shows a range of 6-24% for the wind direction and 10-21% for the wind speed (one minute interval). We used four averaged values for wind speed and direction in OpenFOAM to represent a one-week period of measurements. This implies that many specific wind conditions causing snow redistribution such as periodic turbulent gusts were not represented in the simulations, along with their associated effect on the snow patterns. Aksamit and Pomeroy (2016) showed that these turbulent, short timescale wind structures have a great influence on the particle entrainment and transport. The blowing snow structures are known to be at higher temporal scales but it was not computationally affordable to simulate the complete time period framed by the laser scan measurements. The full directional variability of the wind forcing is likely not well represented by our approach. For the under-represented deposition patches between circles **C** and **D** in particular, Figure 4−IV demonstrates that a stronger flow enables their reproduction by the model at the precise spots. Taking mean wind speed values for forcing tends to damp the intermittent variations of the atmospheric flow, which could have lead to this unmodeled snow accumulation present in the TLS data. Given the various uncertainties linked to the forcing measurements (precipitation, friction velocity) it does not appear relevant to fully simulate the one-week period with snowBedFoam. A quantitative comparison between the results obtained with a fully-resolved simulation and our steady-state rate approach proved to be sufficiently accurate (not shown) in a context of a simplified modelling framework such as the one performed in this work.

2. *Evolution of ice surface structures:* Besides the forcing parameters, some topography-related aspects could explain missing snow features in the model. It is expected that the aeolian redistribution of snow gradually modifies the roughness and topography of the sea ice surface, leading to variations in the flow field and thus in the snow transport processes (Andreas and Claffey, 1995). The invariant mesh that we employ in our simulations neglects this temporal evolution. This could explain why the measurements do not contain the micro-scale distribution patterns found in the north of the modelled ridge: they may have been gradually flattened in reality. Also, the more elongated features of deposition found in the scans could result from the build-up of snow bedforms at the lee of the ridge, which gradually displaced the deposition maximum downwind. Future work will explore the dynamic meshing of the sea ice surface within OpenFOAM as a solution to this modelling limitation.

3. *Atmospheric stability*: Furthermore, the non-consideration of meteorological conditions such as atmospheric stability may have an effect. Various authors have shown that atmospheric stability can have a strong influence on the development of the local near-surface flow field and the associated deposition patterns, together with topography and wind speed (Wang and Huang, 2017; Gerber et al., 2017; Comola et al., 2019). Atmospheric stability influences the vertical motion

of the flow, which can in turn affect the settling of snow particles. The neutrally stable flow imposed in OpenFOAM overlooks this aspect and may introduce inaccuracies in the simulated snow distribution.

4. *Dimensionless parameters - summary:* Overall, we can reasonably assume that the (limited) qualitative disparities in snow distribution between numerical and TLS data are due to a combination of causes. Recent findings of Comola et al. (2019) suggest that different distribution patterns can emerge from different combinations of dimensionless parameters expressing atmospheric stability, particle inertia and length and velocity scales (Froude and Stokes numbers). The simplifications for flow, grain shape, atmospheric conditions and topography imposed by our modelling framework inevitably bring errors; taking this into consideration, the agreement appears to be very satisfying.

Besides the qualitative comparison, we observe that the quantitative performance of our snow model is not optimal. Erosion is underestimated by snowBedFoam both at the ridge and in flatter areas, while deposition is underpredicted in the main snow accumulation patches. We identified sources of error in both the model and measurements that could potentially explain the mismatch.

1. *Precipitation and other measurements*: There are several limitations related to the MOSAiC measurements that could explain the quantitative disagreement with the model. First, an influential source of unreliability lies in the KAZR-derived snowfall estimates. The latter showed a tendency to overestimate precipitation during the MOSAiC campaign (Wagner et al., 2022), which may be the case for event IV. Precisely quantifying this overestimation is challenging, however, as the uncertainty of the KAZR snowfall estimates can be as large as 50% (Matrosov et al., 2022). Exaggerated snowfall rates in the simulations would generate an excess of particles cancelling the erosive effect of the other simulations and leading to more "damped" snow distribution patterns in the combined output. To illustrate, for event IV, simulations run with half of the input rate $I = 0.18$ mm.h$^{-1}$ (compared to $I = 0.39$ mm.h$^{-1}$ in Table 2) resulted in a difference of about 1 kg.m$^{-2}$ in the average snow deposition. This shows that the snowfall rate has a significant impact on the average snow deposition value obtained in the snowBedFoam simulations and should be adjusted with care. However, difficulties arise in obtaining reliable precipitation estimates in the Arctic region (Goodison et al., 1998; Boisvert et al., 2018) and it cannot be determined with certainty which precipitation device was the most accurate during MOSAiC (Wagner et al., 2022; Matrosov et al., 2022). It should be mentioned that the net snow mass loss found in the measurements despite the recorded snowfall suggests that the precipitation overestimation is not the only source of differences. Besides the meteorological forcing, a source of unreliability in the DEMs can originate from the positions of individual points within the scan along with the errors in elevation difference between the two successive surveys: some snow-depth changes could be subject to a measurement error. However, this is expected to be small. In addition, the snow-mass values presented in Figure 6 derive from the multiplication of height data with a constant snow density, which is in reality variable from location to location. Hence, the snow-mass results presented here have errors in representing the actual mass distribution. At last, it is probable that the wind-related measurements brought additional uncertainty in the particle and flow settings of our model; all of these errors combined are expected to have influenced the numerical results.

2. *Temporal variability of the snow cover*: In snowBedFoam, the aerodynamic entrainment of snow is modelled through a fixed threshold value that neglects the temporal changes in the environmental conditions and snowbed properties (metamorphism). The sintering of grains has been identified in the literature as a key component of the equilibrium between erosion and deposition of snow (Blackford, 2007): strong bonds between snow particles might prevent their subsequent erosion and generate stationary bedforms (Filhol and Sturm, 2015). Freshly-fallen snow particles are not bound, thus highly erodible : their transport usually occurs at lower wind speeds than for ice or compacted snow (Guala et al., 2008), resulting in higher mass flux and deposition at the lee side of the ridge. Such conditions likely correspond to the period following the precipitation event on November 11 (Fig. 3), but were not accounted for in the simulations. Moreover, thermal processes such as sublimation were considered to be negligible and this may have impacted the snow mass deposition in the measurements, by lowering the overall snow deposition in flatter areas. These observations suggest that the potential coupling to a snow modelling tool evaluating the snow transport threshold based on meteorological time series such as SNOWPACK (Lehning et al., 1999) could bring great improvement to the results of the CFD model alone.

3. *Spatial heterogeneity of sea ice surface properties:* An uniform snow cover (unvarying parameters over the domain) was considered in our numerical simulations, which contrasts with real snow layers. The latter are usually complex and under natural conditions, they exhibit irregular boundaries and a wide range of grain and bond characteristics when traced laterally (Sturm and Benson, 2004). The same observations apply to the sea ice system, where the snow cover characteristics vary widely from the top of the ridge (non-erodible ice) to flatter areas (weakly bound snow grains). Modelling challenges are here twofold. First, it is difficult to characterize the snow cover properties over space due to a lack of data at the level of detail needed for our simulations. Second, the mathematical parametrization of the snowpack properties has not been fully established yet. Constants such as A (Bagnold's shear stress threshold constant, Equation 10) or $\phi$ (grain cohesion parameter, Equation 13) are a way to include the snowpack properties in snowBedFoam 1.0. but the exact connection between their value and the grain characteristics is still unclear. We are aware that the restricted inclusion of the spatial variability in snow properties may have lowered the accuracy of our results; however, further research is needed to investigate the parametrization of the snow surface properties, which will not be addressed in the present work.

4. *Turbulence scheme:* Furthermore, the use of RANS as a turbulence scheme implies the modelling of averaged flow-fields only, which simplifies the fluctuating nature of snow transport and likely introduces a bias (Groot Zwaaftink et al., 2013). Turbulent eddies intermittently enhance snow erosion and deposition; such atmospheric structures could have incrementally produced sheltered stationary bedforms in the lee of the ridge and lead to the higher accumulation measured by TLS.

5. *Periodic boundary conditions:* At last, the periodic boundary condition imposed at the lateral edges of the domain treats them as if they were physically connected (OpenCFD Ltd, 2019). This approach is usually employed for repeated geometries that only partially apply to sea ice. Figure 1 shows that a relatively flat area precedes the gridded ridge in reality, according to the predominant wind direction (south). The absence of aerodynamic obstacles in the actual terrain

could potentially lead to a higher snow transport flux than in OpenFOAM where the PBCs imply the artificial repetition of the simulated topography on all sides of the numerical domain. In other words, the accumulation of snow particles in reality may occur over vaster areas than in the model where we artificially create successive deposition over the ridges. The higher snow erosion and subsequent deposition observed in the scans could come from larger-scale variations in the horizontal mass flux. In addition, PBCs imply that the eroded particles necessarily deposit somewhere in the numerical domain as they are constantly re-injected through the connected patches. The same applies to injected (precipitation) particles: this explains why there is an important amount of cells (90% of the surface) with a positive snow mass change in the model (Figure 7). Comparing snow measurements on sea ice to a numerical model, which is conservative in terms of mass, highlights the effect of wind redistribution in this environment and can help to understand the snow mass fluxes going in and out of the real system. snowBedFoam 1.0. appears here as an useful tool to assess the snow mass loss (gain) on a given piece of sea ice and to identify potential snow mass sinks (sources). For example, Déry and Tremblay (2004) showed that the total amount of blowing snow loss into the Arctic Ocean may reach between 60 - 100 % but difficulties are encountered in measuring such values; numerical models can help refine these quantitative estimates when comparing their mass-conservative results to real snow distribution measurements.

The limitations identified here highlight the current challenges encountered in the modelling of snow deposition over complex terrain. The spatial variability of the snow cover is caused by physical processes acting at different spatial scales (Mott et al., 2018) and the contribution of each is hardly distinguishable in the field measurements (Gerber et al., 2017). Thus, this field of research relies to a great extent on mathematical modelling (Comola et al., 2019), which inevitably simplifies the complex snow-wind interaction due to both numerical and computational constraints. In the case of sea ice, the inhomogeneous snow distribution mostly results from the wind and precipitation interacting with the snow surface, similarly to what is observed in alpine terrain (Mott and Lehning, 2010). Our snowBedFoam 1.0. model offers the potential to separately simulate the preferential deposition of precipitation (Lehning et al., 2008) and the transport of previously deposited snow, as observable in Figure 4. Thus, the role played by each in the spatial variability of snow deposition can be singled out to better understand its underlying mechanisms.

## 5 Conclusions

In this study we introduce snowBedFoam 1.0., a snow transport model developed on the basis of the standard Lagrangian particle tracking library of the computational fluid dynamics (CFD) software OpenFOAM. We implemented a physics-based splash model, which describes particle mass exchange at the lower boundary. We applied it to simulate the snow accumulation patterns on Arctic sea ice using terrestrial laser scan observations from MOSAiC. To our knowledge this is the first publication on a Eulerian-Lagrangian snow transport model combined with sea ice topographical data from lidar measurements. Qualitatively, results show that most of the snow distribution patterns were accurately captured by the numerical simulations. This demonstrates that small-scale snow transport processes are dominant drivers of the spatial structure of snow distribution over complex terrain. From a quantitative point of view, however, the erosion and deposition were under-represented by the

model in the vicinity of the sea ice ridge while an ubiquitous enhanced snow deposition was simulated in flatter areas. This limited quantitative performance is attributed to various sources such as the CFD numerics, the oversimplification of the real conditions in the model (wind, snowpack properties and related processes) or to the measurements themselves. Further improvements should be incorporated into our modelling approach, notably by coupling snowBedFoam to snowpack models or by including other factors influencing snow transport such as the atmospheric stability. The manifold snow - wind interactions ask for very complex modelling, which is computationally expensive and remains challenging despite the recent developments in computer science.

Although its quantitative performance was not perfect, our model development still represents a significant step towards the accurate modelling of snow deposition on sea ice. This tool could further be used in the assessment of snow mass balance components to improve precipitation estimates through measurements. Overall, considerable progress has been made towards the small-scale study of snow distribution as the model may be further applied to other topographies, with the advantage of incorporating elevation data stemming from laser scans.

*Code and data availability.* The scripts of the snowBedFoam 1.0. model are available on the EnviDat data repository of the Swiss Federal Institute WSL (Hames et al., 2021). The raw Met City meteorological tower data are available at the Arctic Data Center (Cox et al., 2021). The KAZR data from which the snowfall was derived is available at the ARM data archive (Lindenmaier et al., 2019). The Terrestrial Laser Scan data processed by D.C.-S. can be found on the Arctic Data Center repository (Clemens-Sewall, 2021).

*Author contributions.* O.H. took the lead in writing the manuscript but declares joint first-authorship with M.J. Both M.J. and O.H. implemented the snow transport model in OpenFOAM and ran the snowBedFoam 1.0. simulations. D. C.-S. and C. P. were responsible for the planning of the TLS campaign and provided the scanner and accessories. D. C.-S. processed the TLS point clouds to generate the sea ice surface position data. I. R. and D. N. W. conducted the scans and D. N. W. measured and processed snow density, TLS and KAZR data. He and M.S. were in charge of the planning and coordination with the MOSAiC organization. M.D.S. and his team measured and processed the meteorological data employed for the model forcing, which we published in its preliminary version. M.D.S. was also the principal investigator for the ARM involvement in MOSAiC (KAZR data). M.L. encouraged O.H. and M.J. to investigate the snow transport on sea ice, gave regular input on the implementation of physical processes and supervised the work. Besides, he helped design together with M.S. the numerical set-up and relevant MOSAIC field deployments.

*Competing interests.* The authors declare that they have no conflict of interest.

*Acknowledgements.* The authors thank Varun Sharma and Daniela Brito Melo for their help in better understanding their Large Eddy Simulation-Lagrangian Particle Tracking FORTRAN code as well as sharing their knowledge about fluid dynamics. Franziska Gerber and

Robert Kenner are also thanked for their valuable help and guidance in the processing of the TLS data. Finally, the authors thank all MOSAiC

participants for contributing to the paper in one way or another: the MOSAiC logistics- and data team, the ship's personnel, members from all science teams on board, the ARM crew, the MOSAiC project team and the project board.

The data used in this article was produced as part of the international Multidisciplinary drifting Observatory for the Study of the Arctic Climate (MOSAiC) with the tag MOSAiC20192020 and the Polarstern expedition number AWI_PS122_00. Precipitation data were obtained from the Atmospheric Radiation Measurement (ARM) user facility, a U.S. Department of Energy (DOE) Office of Science user facility man-

aged by the Biological and Environmental Research Program. M.D.S. was supported by the DOE Atmospheric System Research Program (DE-SC0019251, DE-SC0021341) and the National Science Foundation (OPP-1724551).

This project is supported by the Swiss National Science Foundation-SNF, grant number 200021E-160667 and 200020-179130.

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

900

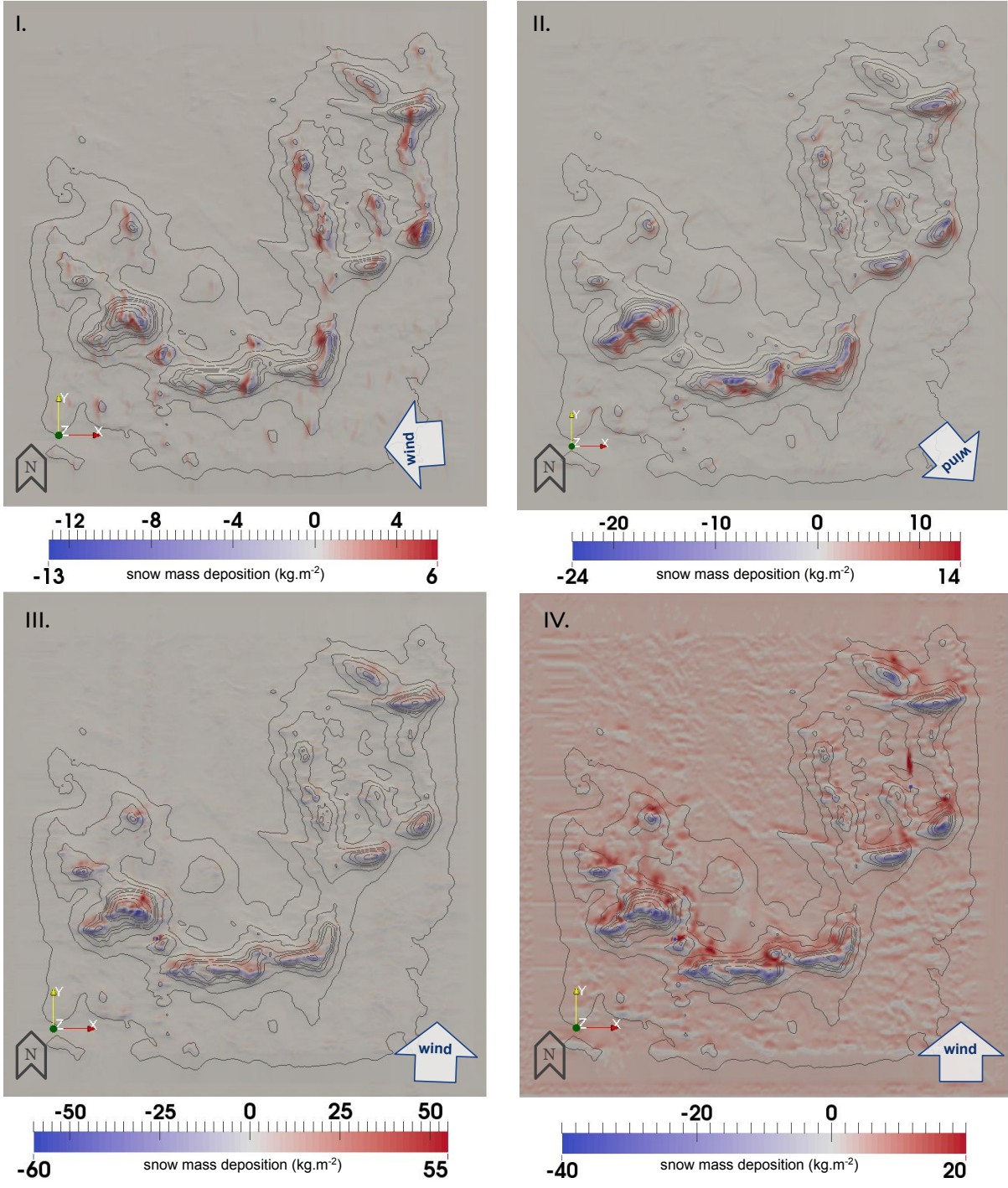

**Figure 4.** Extrapolated snow deposition results (kg.m$^{-2}$) for the four atmospheric events (I−IV) identified in the period between the two successive laser scans. The arrows in the bottom right corner indicate the direction of the fluid forcing.

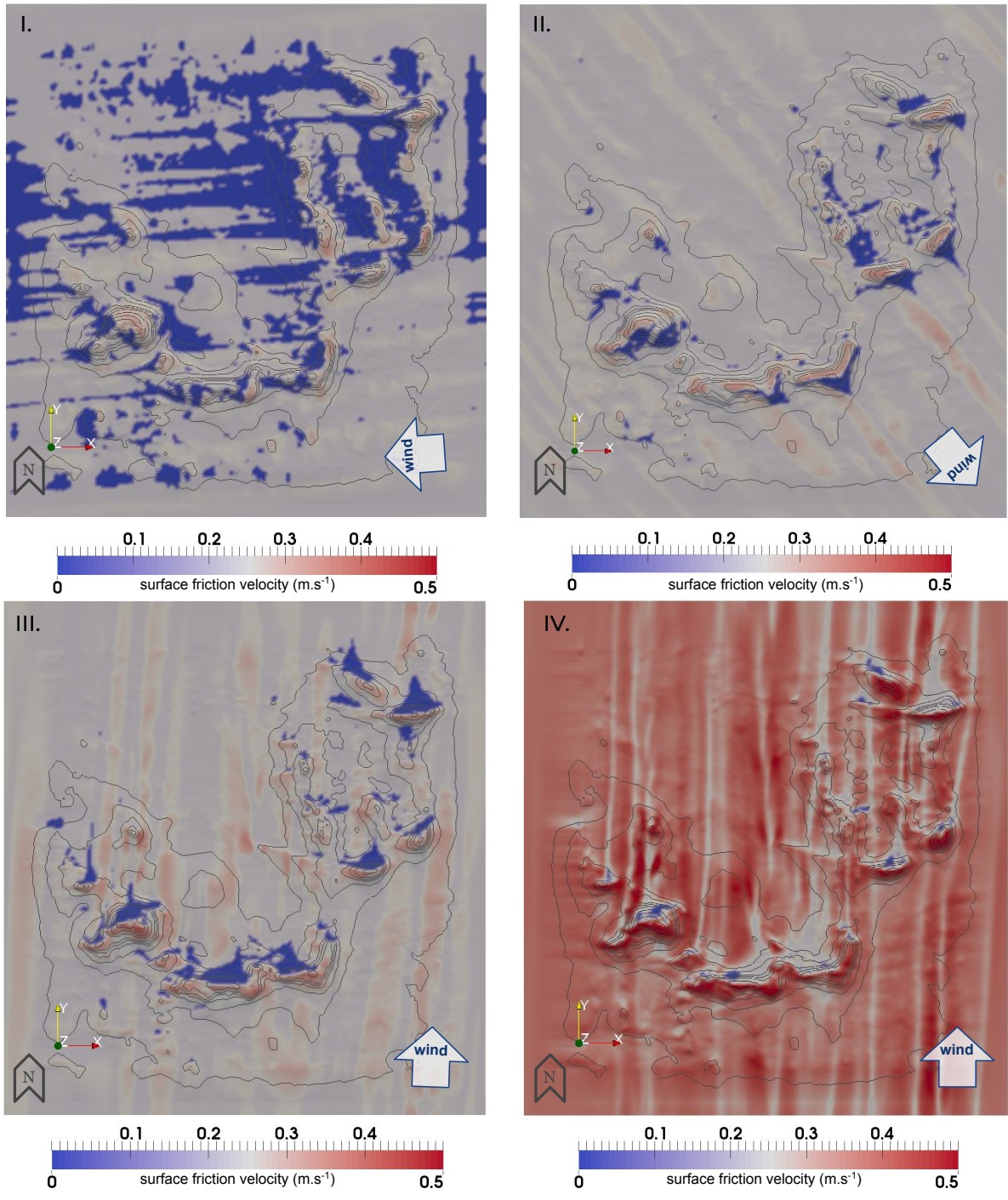

**Figure 5.** Friction velocity results (m.s$^{-1}$) for the four atmospheric events (I–IV) identified in the period between the two successive laser scans. The arrows in the bottom right corner indicate the direction of the fluid forcing.

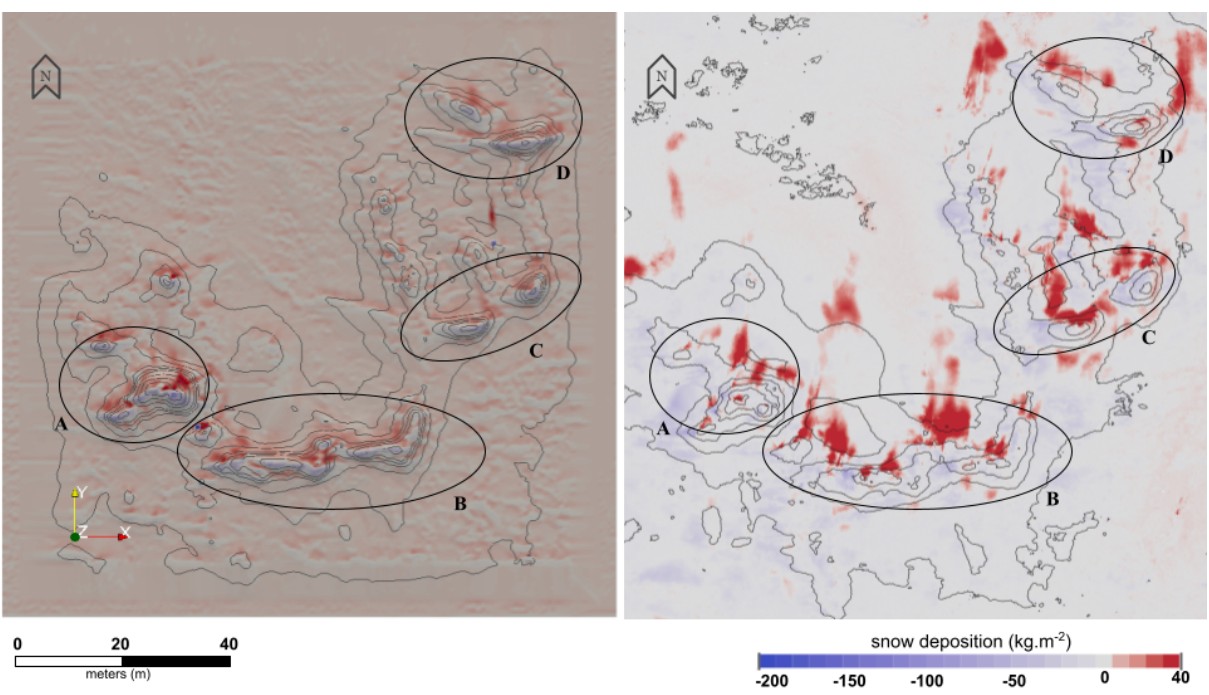

**Figure 6. Left:** numerical snow deposition patterns (kg.m$^{-2}$) after the combination of the extrapolated snowBedFoam 1.0. simulation results from the four individual atmospheric events (I−IV). **Right:** snow-mass differences obtained by differencing two successive elevation models measured during MOSAiC.

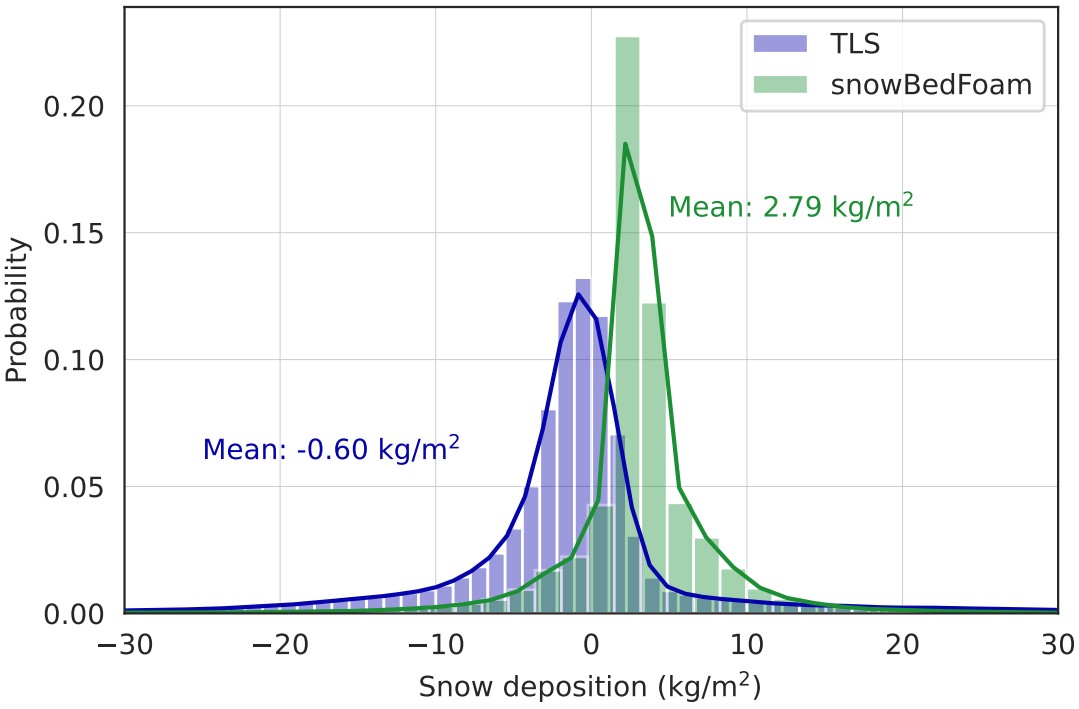

**Figure 7.** Statistical distribution of snow deposition (kg.m$^{-2}$) for the combined extrapolated snowBedFoam 1.0. simulation results (green) and the terrestrial laser scan measurements (blue). The average snow mass change values for TLS and OpenFOAM are displayed in the corresponding colours.