# Peer review of "Modelling the small-scale deposition of snow onto structured Arctic sea ice during a MOSAiC storm using snowBedFoam 1.0."

_Geoscientific Model Development, 2021_

## Author Response (AR1)

**Responses to Referee 1 for the manuscript:**

**"Modelling the small-scale deposition of snow onto structured Arctic sea ice during a MOSAiC storm using snowBedFoam 1.0."**

April 7, 2022

**1 General answer**

The present document constitutes the author's response to the first referee, who posted his/her comments on October 5, 2021 in the open discussion of our GMD manuscript. We thank the reviewer for his/her thoughtful comments and efforts towards improving our manuscript as well as for the expertise provided. The numerous references that he/she suggested were especially helpful to further develop our work. The references to the lines in the revised manuscript are included within square brackets, such as [L00]. In the author's track-changes file, the changes based on the comments of the 1st referee (posted on October 5th, 2021) are highlighted in blue color. The changes in green color are related to the remarks of the second referee (posted on January 3, 2022) and the violet color are the changes related to the comments of all referees. The additional changes added by the authors appear in gray. Note that some values for the forcing parameters have been changed, based on the most recently processed meteorological data (Table 1 of the revised manuscript). In the present document, the remarks of the referee are in black color and the author comments are in blue.

**2 Major comments**

For the sake of clarity, we chose to divide the major comments in sub-sections and named them after their content.

**2.1 Snow sublimation**

"My first concern is the lack of consideration for a sublimation flux. Sublimation of blowing snow and of snow packs has been identified as a major contributor to mass loss in many environments (Mott, 2018). The authors do not describe the humidity of this location. Indeed, perhaps it is sufficiently humid that such an approximation is warranted. However, this is never described nor justified in anyway. I see that the MOSAiC companion paper Wagner (2021; TCD) notes low sublimation fluxes for this period, perhaps 6%. This seems derived from modelling studies and not observations. The simulation period is short and perhaps sublimation is negligible. However, that has not been demonstrated and to wave it away, especially when the model is noted to have deviations from observations, seems problematic. I would like to see a sublimation sink added and

evidence that, in a distributed context that this process is indeed negligible."

Thermal processes such as sublimation (surface and blowing snow) are not taken into account within snowBedFoam, even though they may be of significant importance for the snow mass distribution depending on the spatial and temporal scales of interest (Mott, Vionnet, and Grünewald 2018). The estimation of mass loss due to snow sublimation at the surface and by blowing snow is shown to be challenging (Mott, Vionnet, and Grünewald 2018) and to our knowledge, direct field measurements for snow sublimation fluxes are not yet available in the literature for Central Arctic. For our specific case, we assume the sublimation flux to be negligible for the considered period of time (i.e. November 6 to 14, 2019) and location (Central Arctic) based on two main modelling studies. First, we refer to the earlier work of Chung, Bélair, and Mailhot 2011 who have suggested that Arctic blowing snow sublimation may represent a mass sink of about 6% of the total cumulative snowfall. This result is based on simulations by the complex snow model PIEKTUK (e.g. Leonard and Maksym 2011) which were ran for a period of 324 days between November 1997 to September 1998. The second, more recent work of Webster et al. 2021 analyzes the results generated by different versions of the Community Earth System Model (CESM) simulating snow on Arctic sea ice. The simulated snow surface sublimation fluxes appear to be very low on average ($< 0.004$ cm/day) in November for the 2000s. These studies support our assumption that the energy available in the atmosphere during polar night in the Arctic is not sufficient to drive sublimation processes. We are aware that this assumption is not valid for all times and locations and future work should imply the addition of snow sublimation processes in snowBedFoam as they can be a substantial component of the snow mass balance. We comment on the sublimation sink in several places of the revised manuscript such as "Only pure mechanical fluid-particle interactions are considered here, thus we distinctively evaluate the impact of the horizontal snow transport on the sea ice snow mass balance at a given location. Thermal processes such as the sublimation of blowing snow and snow at the surface, although having a big role in the snow mass budget at certain spatio-temporal scales, are assumed to be negligible given the time period and location of interest (Chung et al. (2011); Webster et al. (2021))" [L91]. Furthermore, a detailed explanation of our hypothesis (similar to the lines above) is given in Section 2.4. Modelling assumptions of the revised manuscript [L373]. The topic is also further discussed in Section 4 [L542].

**2.2  Snow transport references**

"The authors extensively cite Bagnold 1941 without ever noting that it is a sand transport study. I would encourage the authors to cite the vast body of literature surrounding the early work of adopting this work for blowing snow such as, e.g., Budd, 1966; Schmidt, 1982; and Pomeroy, 1990."

This suggestion has been taken into account in the revised manuscript (see section 2.2.2 in particular [L225, L237]). More diverse, snow-specific sources have been added to the description of snow transport and more thorough explanations on Bagnold's work were included in the text. We thank the reviewer for the suggested blowing snow references.

**2.3  Low-velocity streamwise streaks**

"My second concern is shown in Fig 5. Specifically the long, low velocity streaks of u*. Ivanell (2018) and Wagenbrenner (2019) have a discussion of similar streaks in RANS models. These areas may have substantial impacts on blowing snow simulations and deserve attention. I would like to see, at a minimum, a description and placement in the literature of these features and if the authors

think they are real. Wagenbrenner (2019) identifies that they are somewhat dependent upon the upwinding scheme used. Do the authors think that is the case here?"

To support our explanations in this sub-section, a zoom on Figure 5-IV of the manuscript is provided in Figure 1, which should improve the visualization of the low-velocity streamwise streaks analyzed hereafter. We use the references provided by the reviewer as a base for the discussion.

In their micro-scale model comparison study, Ivanell et al. 2018 present numerical results with similar streaks to the ones obtained with our snowBedFoam model and support that such patterns correlate with topographical features. They have also observed streaks for simulations ran without topography, however, but did not provide any justification for such observations. Similarly, in their Large-Eddy Simulation (LES) runs on flat terrain, Groot Zwaaftink et al. 2014 observe low-shear stress structures in their modelled fields. Wagenbrenner et al. 2019 explain more extensively the potential source of similar streaks appearing in their wind model results; in reference to experimental and observational evidence, they suggest that these streamwise low-velocity structures are real and induced by the terrain. Their hypothesis is that counter-rotating vertices forming in the wake region of the mean flow and converging after the topography create such streaks. Hesp and Smyth 2017 show that the shape and aspect ratio of the topography have a direct impact on the length of these streaks and that the mean flow is typically characterised by two symmetrically opposed vortices within the flow separation wake zone.

Based on the literature and our own modelling experience, we suggest that the long, low-velocity streaks shown in Figure 5 of the manuscript are real and induced by the topography, but that the choice of boundary conditions and discretization schemes may have impacted their shape and extent. Similarly to Hesp and Smyth 2017, we found counter-rotating vertices in the wake region of the sea ice ridge (Figure 2, below), which could explain the occurrence of the streaks following downwind in the domain. The vertical blue line in Figure 1 indicates the location where the rotating streamlines were sampled. Figure 2 shows the occurrence of mean flow eddies directly in the lee side of the topography, which are then deviated from their original X-position and continue along a trajectory showing low friction velocity at the surface (left side of Figure 2). The right side of Figure 2 shows a surface mesh without colour to ease the visualization of the eddies. The complex ridge topography induces turbulence in its vicinity and the decelerated flows in the wake region may explain the elongated low-velocity zones found in our simulations. Regarding the influence of the numerical settings on these streaks, we used a second-order linear upwind discretization scheme for the convection term of flow velocity, which has shown to produce broader and longer low-velocity streaks compared to other discretization schemes (Wagenbrenner et al. 2019). Therefore, our choice in the discretization scheme may have impacted these structures. Besides, another influencing numerical setting is the type of boundary conditions employed. Periodic boundary conditions imply that when an object passes through one side of the domain, it re-appears on the opposite side with the same velocity. This means that the lower velocity found down-wind of the topography is translated directly to the beginning of the domain, thus creating a zone of lower-velocity flow before the ridge, which is continued after the topography due to turbulence. This is not really problematic because it can represent the reality of sea ice, where the upwind topography has an impact on the flow field reaching the ridges located downwind.

In summary, we believe that these streaks are natural but may be over-represented in our simulations due to the boundary conditions and the discretization scheme employed in our work. It is part of the numerical uncertainties that can be hardly avoided and that are difficult to quantify, as there is very little guidance in terms of the realistic representation of these streamwise flow features (Wagenbrenner et al. 2019). A detailed explanation was added to the revised manuscript, in Section 3.1. Snow distribution patterns per event [L410].

[Figure]

Figure 1: Friction velocity results $(\mathrm{m.s^{-1}})$ for simulation IV. The vertical line represents the position where the turbulent streamlines shown in Figure 2 were sampled.

**2.4 Sensitivity analysis for model parameters**

**2.4.1 Meteorological forcing**

"This brings me to my last concern – a lack of uncertainty analysis. I would like to see the authors quantify the impact of any meteorological forcing, e.g., their input precipitation and the values in Table 2. Certain values can have massive impacts, e.g., friction velocity threshold, and it would be useful to understand how sensitive the model is to these parameters."

Regarding the quantification of uncertainty in connection with the input precipitation, we supplement our answer with additional snowfall measurements taken during the MOSAiC campaign for the period November 5 to November 15, 2019 (Figure 3). They cover the time period between the two terrestrial laser scans investigated in our study. The top of Figure 3 shows the precipitation rates estimated by 5 different sources (4 instruments, 1 meteorological model). The values represent the Snow Water Equivalent (SWE) of snowfall. PWD22s correspond to Vaisala Present Weather Detector 22 optical sensors, Pluvio2 is an OTT Pluvio[2] pluviometer, KAZR is the radar-derived data described in the manuscript and ERA5 corresponds to the mean snowfall rates calculated with the ERA5 model (Hersbach et al. 2020) from the European Centre for Medium-Range Weather Forecasts (ECMWF). More details about the measuring principle and performance

[Figure]

Figure 2: Zoom on the vertices found in the lee of a sea-ice pressure ridge section. **Left:** the colors at the surface represent the friction velocity and correspond to the color bar legend displayed in Figure 1. The color legend for flow streamlines is given in the bottom left corner. **Right:** the surface is colorless to improve the visualization of the flow streamlines.

of the instruments can be found in Wagner et al. 2021.

During the MOSAiC campaign, the KAZR data has been identified as the possible upper cumulative limit of the precipitation range while the PWD22 sensors showed a tendency to under-estimate solid precipitation (Wagner et al. 2021), which was previously observed in other studies (Wong 2012). However, it appears in Figure 3 that the PWD22 device with the lowest cumulative snowfall (located on the Polarstern ship, yellow curve) was deficient during the event of interest on November 11, 2019. There is a limited amount of values available for this period. Therefore, we take the precipitation rates measured by the Pluvio2 sensor, which recorded the second lowest cumulated snowfall for the observed period. The precipitation rates measured by Pluvio2 during the snowfall event simulated with snowBedFoam are about half of the ones measured by the KAZR radar (0.18 vs 0.39 mm.h$^{-1}$ on average). To evaluate the impact of this difference on the simulation results, we ran an additional simulation with the same settings as Event IV of the manuscript, but with the Pluvio2 precipitation rate as input data, namely $I$=0.18 mm.h$^{-1}$. Then, similarly to the other simulations, the results after the 7 hours duration snowfall were computed using the snow deposition and erosion rates after 1000 s of simulation (see Section 2.4.3 of the present document for a fully detailed description of the method). The difference in snow mass distribution results between the higher-envelope and lower-envelope input precipitation values are displayed in Figure 4. As expected, the snow mass deposition is higher for the simulation with the highest input pre-cipitation rate (KAZR radar). The erosion at the ridge also appears to be higher, likely due to the impact of the precipitation particles at the snowbed surface. The quantitative values given in Table 1 further support these observations. Interestingly, using the Pluvio2 measurements as input precipitation data for simulatino IV would actually result in a shift of the probability distribution peak of the modelled snow mass deposition to the left, thus closer to the measured values (see Figure 7 of the manuscript). However, it is very difficult to assess which precipitation measurement is the most accurate at a given time because the actual "true" precipitation value is not known. As

[Figure]

Figure 3: Precipitation rates (top) and cumulated precipitation values (bottom) in Snow Water Equivalent (SWE) estimated during the MOSAiC campaign with radar (KAZR), optical devices (PWD22), pluviometer (Pluvio2) and meteorological model (ERA5).

an illustration, uncertainties up to 50% were observed for the KAZR-derived snowfall rates (Matrosov, Shupe, and Uttal 2022). We only have access to a range of precipitation values. The results

Table 1: Quantitative details for the snowBedFoam simulation results computed with Pluvio2 and KAZR measurements as input precipitation data, respectively.

| Device | Snow deposition | | | Snow erosion | | |
|---|---|---|---|---|---|---|
| | mean $\mathrm{kg.m^{-2}}$ | maximum $\mathrm{kg.m^{-2}}$ | area $\%$ | mean $\mathrm{kg.m^{-2}}$ | maximum $\mathrm{kg.m^{-2}}$ | area $\%$ |
| Pluvio2 | 1.6 | 18.2 | 84.2 | -2.5 | -43.93 | 15.7 |
| KAZR | 2.5 | 18.9 | 93.4 | -3.9 | -42.7 | 6.6 |

of this sensitivity analysis have been briefly mentioned in the discussion of the revised manuscript (paragraph *1. Precipitation and other measurements* [L519]). The impact of friction velocity as meteorological forcing is discussed in Section 2.5.

**2.4.2 Shear stress threshold**

Besides the precipitation rate, whose impact on the results was evaluated above, the values shown in Table 2 are commonly used in the snow literature and should give reasonable snow threshold evaluations. The friction velocity threshold defined in Equation 10 of the manuscript is based on the Bagnold's shear stress formulation, which has proven to give satisfying results empirically with an A parameter value of 0.18 (Clifton, Rüedi, and Michael Lehning 2006). A sensitivity analysis previously conducted with the LES-LSM model (Groot Zwaaftink et al. 2014; Comola et al. 2019; Sharma, Comola, and M. Lehning 2018) having similar formulations for the snowbed-grains interactions showed that varying the A constant for a range of values from 0.1 to 0.3 only subtly affected the total snow mass flux in the domain. Regarding the effect of the particle diameter involved in the shear stress threshold expression on the results, we refer to the work of Melo et al. 2022 who studied the impact of mean grain size on saltation fluxes using LES-LSM simulations. Comparably to other snow transport studies (e.g. Sørensen 2004), the authors did not find an explicit variation of the snow mass flux with the particle mean diameter. This has been included in the revised manuscript, Section 2.3.4. [L365].

**2.4.3 Simulation time**

"It is not 100% clear to me how the simulations were done. It seems to be developing a steady-state (?) simulation of 1000 s at which point the wind speed is reduced? How sensitive are the results to this (spin up?) 1000 s period?"

In our approach, the simulation time of 1000 s was determined through the analysis of the total snow mass and average friction velocity in the numerical domain over time. We illustrate the subsequent explanation with simulation output from the first event (I) in Figure 5. It is observed that the total mass in the domain oscillates around a value of 0.3 kg, which translates a flow-particle system in equilibrium. We chose a simulation time of 1000 s to make sure that a steady-state saltation was reached. Figure 6 shows the evolution of the average surface friction velocity in the domain over time. The computation of the surface friction velocity is performed in snowBedFoam through the Lagrangian module, which implies that there are no computational results for the first 100 s of simulation (only the Eulerian phase is modelled to let the flow fully develop). It can be seen that after a significant drop in friction velocity due to the introduction of particles, the

[Figure]

Figure 4: Difference in snow mass deposition (kg.m$^{-2}$) between the results obtained with KAZR radar (higher envelope of snowfall measurements) and Pluvio2 (lower envelope of snowfall measurements) input precipitation data.

velocity quickly stabilizes (after 200 s) around 0.235 m.s$^{-1}$. The analysis of the total integrated mass flux and surface friction velocity over time manifest a system having reached an equilibrium state, which is how the simulation period was chosen. Using the 1000 s of the simulation, we derive the deposition and erosion rates in each cell from the snow distribution results between 600 s and 1000 s (steady-state), which we then multiply by the total duration of the simulated event. Deriving the rates based on the 400 s or 500 s simulation output gives similar results. The method is explained in the revised manuscript, Section 2.3.4 [L349].

[Figure]

Figure 5: Total snow mass in the system (kg) over simulation time for event I.

**2.5   Evaluation of the averaged wind method for a several-hour period**

"The large temporal periods over which the simulation is run from mean values of wind is concerning or at least requires further discussion. The temporal scales that impact blowing snow are quite small, < 1 s (Aksamit & Pomeroy, 2016, 2018), although at 15 m to 1 h r scales, mean shear-stress models tend to be successful. However it is not clear how successful a many-hour mean wind structure is for representing these features. To me it seems a mis-match to run a sub-metre spatial model, but drive it with many-hour mean windflow that we know doesn't represent any of the wind structure known to drive blowing snow events."

The temporal decomposition into dominating phases of velocity has first been introduced by Mott and Michael Lehning 2010 based on wind fields generated by ARPS runs and has proven successful at a scale of decimeters. We developed our approach of decomposing a whole week into 4 dominating wind events based on their work. We would like to point out that our RANS approach only approximates the temporal dynamics observed in turbulent flows.

For a given dominant event, based on the temporal behaviour of the total particle mass (Figure 5) and averaged friction velocity (Figure 6) in the domain, we assume that the erosion and deposition rates at the sea-ice surface stay constant (equilibrium state) and that subsequent snow distribution values at a given time can be derived from the results computed at the beginning of the simulation, by multiplying the rates at each cell (kg.m$^{-2}$.s$^{-1}$) by the duration of interest (s). To verify whether the use of an averaged wind speed is successful in approximating the snow distribution patterns yielded after several hours of fluctuating flow, we ran a fully-resolved simulation for event I, which occurred on November 6 2019, from 12:00 to 23:00. By "fully-resolved simulation", we mean that the particle-flow dynamics were simulated for a total of 12 hours, with the flow driving force being

[Figure]

Figure 6: Average friction velocity in the domain over simulation time for event I (starting at 100 s).

changed every 3600 s using the friction velocity and wind direction values measured during MOSAiC for the period of event I (see the measurement values in Figure 3 of the manuscript).

It takes approximately 1.5 day of real time to fully simulate one hour (3600 s) of flow-particle interactions. The 12 hour-long simulation of event I presented here took almost three weeks to be achieved. These values emphasize that the computational burden of a fully-resolved one week period of simulation is not affordable. Figure 7 (bottom) shows the results of the total mass in the system, for the whole simulation time (43200 s). The friction velocity used to determine the flow driving force is represented in the top of the figure through the orange color bar. A clear positive correlation appears between the latter and the total mass in the system. We can observe from the figure that the friction velocity has an important impact on the mass in the system, which is highly non-linear over the simulation time. Interestingly, friction velocity values close to 0.2 m.s$^{-1}$ make the mass in the system drop significantly towards zero while a 10% increase from 0.27 to 0.3 m.s$-1$ (at timestep = 14 400 s) generates almost the double of the mass in the system. This means that small uncertainties in the friction velocity measurements can lead to quite different estimated mass fluxes. In the friction velocity measurements at 2 m derived from the towers at 2 m, 6 m and 10 m during MOSAiC, the ratio between the standard deviation of the measurements from the 3 towers and their average was about 6%. Note that these values have slightly changed since the submission of the manuscript (further post-processing was performed by the MOSAiC atmosphere surface energy flux team such as filters etc). We updated the simulation results presented in the revised manuscript based on the most recent friction velocity estimates (see Table 1, page 14).

The snow distribution results computed with the snow deposition and erosion rates obtained with an average mean-flow of 0.27 m.s$^{-1}$ (multiplied by a 43200 s duration) after 1000 s and the fully-resolved simulation (changing flow, every 3600 s) are compared in Figure 8. Qualitatively,

[Figure]

Figure 7: Friction velocity used for flow driving force (top) and total mass flux in the domain (bottom) over time for the fully-resolved simulation of event I. The orange dots indicate a change in the flow driving force (strength, direction) every 3600 s.

the patterns appear to be almost identical. The quantitative difference between the two snow distribution patterns is shown in Figure 9. The values for snow distribution estimated around the ridge are either over- or under-estimated with the rate-derived results: no clear trend is seen in the difference values, which show unstructured patterns. The difference between the rate-derived and fully-resolved simulations is significant, but it is small considering the high degree of non-linearity of the process. These results show that the qualitative patterns can be accurately captured with a mean wind-flow simulation and with an acceptable quantitative error, despite highly complex processes. We agree that our approach may seem like a mis-match between the observed temporal and spatial scales but it is still a good compromise to obtain satisfying results with a limited computational effort. Moreover, the uncertainties related to the meteorological forcing and numerical framework limit one's interest in running a fully-resolved simulation for a one-week period. For the purpose of exploring the capability of our snowBedFoam model in reproducing the snow distribution patterns on sea ice, we consider that our mean wind-flow approach is reasonable. A reference

[Figure]

Figure 8: Snow distribution results derived from the snow erosion/deposition rates after 1000 s (left) and obtained after a fully-resolved simulation.

to this comparison was added to the revised manuscript in Section 4, paragraph 1.*Simplification of wind transport* [L483].

**3 Additional Comments**

**3.1 Page 1**

1. L1 > "In these" This sentence is not clear

   Changed to "In these polar regions covered by sea ice" [L2].

2. L2 > has a substantial effect. On what?

   Changed to "is relatively strong due to the absence of obstructions" [L2].

3. L3 > a large part of the snow. Of the snow mass? Clarify what this means.

   Changed to "deposited snow mass" [L3].

4. L3 > which complicates precipitation estimates. Oh so this is precip undercatch?

   There are precipitation measurements available close to the simulated region but it is not the purpose of this paper to do a direct comparison with them. This sentence was meant to be more general. Further explanations were added to make it more clear: "which complicates estimates for precipitation hardly distinguishable from blowing or drifting snow" [L3].

[Figure]

Figure 9: Difference in the snow distribution patterns $(kg.m^{-2})$ between the results derived from the mean-flow simulation and the fully-resolved simulation for Event I.

5. L5 > to remove these uncertainties. It's not clear what these are. Do you mean surface precip?

   We meant incertitude about the snow mass balance on sea ice and the relative importance of the erosion/deposition of snow grains at a given location. The sentence was changed to: "remove the snow mass balance uncertainties (i.e. snow transport contribution) in the Arctic environment" [L6].

6. L8 > on a piece of MOSAiC sea ice. MOSAiC domain?

   It adds clarity to the sentence, indeed. Changed to "in a MOSAiC sea ice domain" [L9].

7. L9 > terrestrial laser scan observations?

   Yes, it has been modified [L8] but to avoid the repetition in the subsequent sentence we changed "terrestrial laser scans" to "scanner measurements" [L11].

8. L11 > Could help to better constrain precipitation estimates. This is not clear from the abstract why this is the case and it remains unanswered.

   Hopefully the above modifications help with this issue.

9. L16 > seems to respond. Word choice for respond.

Replaced by "be impacted" such as: "be strongly impacted by climate change" [L18].

10. L16 > very. This was changed, see sentence above.

    17 > an important reduction. Significant reduction? Yes, this works. It was changed in the revised manuscript [L19].

**3.2 Page 2**

1. L 20 > : the i. Start new sentence. Changed [L23].

2. L29 > Also, Arctic precipitation estimates could be significantly improved by an accurate assessment of the snow deposition on sea ice. How? It is well known that snow depth on the ground doesn't equal snow fall due to sublimation.

   A more developed answer to the question of sublimation can be found in subsection 2.1. We added more information in the Introduction of the manuscript to make sure to be understood by the reader: "Also, the strong winds encountered in the Arctic environment lead to large uncertainties in both model projections and measurements of precipitation (Goodison et al., 1998; Wong, 2012; Boisvert et al., 2018), partly due to blowing snow being falsely detected as precipitation by snowfall sensors (Sugiura et al., 2003). Arctic precipitation estimates could be significantly improved by an accurate assessment of the snow transport and redistribution on sea ice, assuming that the other snow mass35 sink terms (e.g. sublimation/condensation and runoff) are known or negligible. Outside the melting season (polar night and adjacent months), the erosion of snow has been identified as the largest sink term and may cause up to a 50 total precipitated snow mass on sea ice (Leonard and Maksym, 2011). In comparison, processes such as sublimation, melt or condensation have shown to play small roles in the snow mass budget for the same period (Webster et al., 2021)" [L32]. We hope this section is more clear now.

3. L33 > Connecting the snow mass balance to snowfall. What about sublimation?

   The answer above also relates to this comment.

4. L35 > that gets. That is. Changed [L44].

5. L38 > the influence of each of these processes. Influence where? Certainly these have been well constrained in many studies.

   By influence we meant contribution of the various processes to the snow mass balance in Arctic regions. The sentence has been changed in the manuscript to: "the contribution of each of these processes to the Arctic snow mass balance, however, is not yet fully resolved" [L45]. Also, references to studies quantitatively constraining those processes were added.

6. L45 > To our knowledge, such spatial observations of snow deposition are non-existent in the literature for Arctic sea ice. Is this not a summary of Trujillo 2016? Please clarify.

   The work of Trujillo et al. 2016 is about sea ice in Antarctica. We wanted to refer to the sea in the Northern Hemisphere, in particular.

7. L 54 > which we perform in this study. Changed to "which we achieve in the present work" [L62] to avoid any confusion about which study (theirs or ours) we are actually referring to.

**3.3 Page 3**

1. L54 > spatial variability of snow deposition around complex terrain. To be clear you mean blowing snow transport?

   We refer here to the snow deposition patterns at small scales, which are subsequent to the snow transport processes (saltation and suspension). The text now describes both the transport processes and the deposition patters they induce: "Generally speaking, the wind-induced snow transport processes near the ground (saltation) and at higher elevations such as suspension and preferential deposition (Lehning et al., 2008) are dominant drivers for the spatial variability of snow distribution at small scale (few meters to hundreds of meters) and shape the snow deposition patterns across various environments (Mott et al., 2018)" [L63].

2. L54 > is not yet fully understood Grünewald et al., 2010). Is there a more recent reference ?

   More recent references were added to the text, and the whole sentence was re-structured [L65].

3. L 55 > Multiple model approaches exist that try to describe it. If the previous section is indeed blowing snow then this is missing many references. But what is 'it' referring to?

   We are here talking about high-resolution CFD models for snow-atmosphere-terrain interaction. We clarified it in the manuscript with the following sentence: "Multiple studies describe and try to reproduce those small-scale snow deposition patterns at high resolution through the modelling of snow transport processes with detailed terrain-flow-particles interactions (e.g., Gauer, 2001; Mott and Lehning, 2010; Groot Zwaaftink et al., 2014; Wang and Huang, 2017)." [L66]

4. L 63 > on a piece of sea ice. Is this the technical definition? Please be more specific.

   Piece of sea ice is indeed a bit vague. We replaced the terms with: "in a numerical domain containing a second-year sea ice topography with typical pressure ridges." [L76]

5. L 63 > Several data sets. Datasets of what?

   These data sets are described in the sentences right after [L78].

6. L 63 > MOSAiC (Multidisciplinary Drifting Observatory for the Study of Arctic Climate). Define this earlier.

   The acronym is now spelled out in the abstract [L9], and a second time for its first appearance in the body of the manuscript [L77].

7. L 65 > The first one. Data set.

   Changed to: "The first data set consists of terrestrial laser scans (TLS)" [L78]

8. L 79 > This article. Consider "manuscript". It is more accurate, indeed. Changed [L102].

9. L80 > second stage, "Section". Changed [L103].

10. L80 > I would like to see scientific questions and hypothesis testing!

    A section about our goals and expectations has been added to the manuscript. Here is an extract: "Previous studies demonstrated that the small-scale snow transport processes mainly drive the spatial structure of snow distribution (Gerber et al., 2018), and it can be expected that a strict Eulerian-Lagrangian snow transport model can reproduce the snow

distribution patterns on sea ice qualitatively. An accurate quantitative evaluation of the snow mass distribution is less likely, however, given the measurement uncertainties and modelling simplifications implied by our numerical framework." Please, refer to the manuscript to see the modifications as a whole [L95].

**3.4 Page 4**

1. L87 > data sets employed here. "Used" here.

   Changed [L110].

2. L 97 > that were successively operated. Do you mean observed?

   We meant here the "scans" as measurements, not the results. "operated" was replaced with "measured" in the manuscript [L121].

3. L 101 > was placed as high as possible. How high is this?

   We added the following details: "(approx. 2.7 m above level surface)" [L128].

4. L103 > were recorded relatively. Relative.

   We replaced the term by "within" for more clarity [L129].

5. L103 > Details about the use of TLS for sea ice measurement. Does that describe this dataset or in general? Please clarify.

   We meant in general. We added the term "General details" to be more precise [L129].

6. L 104 > Several corrections. To the interpolated or the raw point clouds?

   We meant to the raw point clouds. The order of the sentences in the text has been changed to improve clarity: "Before its use, the raw point cloud was post-processed with the RiSCAN PRO v2.10. (RIEGL, 2020) software. Several corrections were made (...)" [L130].

7. L 104 > and interpolated. Spatially? What method?

   Spatially interpolated with the TIN Method. Specified in the text: "The processed point clouds were then spatially interpolated into digital snow surfaces using the Triangulated Irregular Network (TIN) (or Delaunay) interpolation method, available in the QGIS open-source software (QGIS.org, 2022)." [L132]

8. L 105 > is a last step, the point clouds were aggregated. Is this the QGIS step?

   The QGIS steps include the interpolation of the point clouds (see above) and the export into grid cells. We specified it for both in the manuscript: "Finally, the interpolated data was exported at grid scales between 20 cm to 1 m with QGIS." [L134]

9. L 110 > with a constant snow density value of 210 kg.m$^{-3}$. Was this an in situ observation on the ground or fresh snowfall?

   This is derived from on-ground snow density measurements conducted in the area where the scans were taken. We added in the manuscript as follows: "This density value is derived from on-ground bulk measurements conducted in situ with an ETH tube and a snow density cutter (Haberkorn, 2019)." [L139]

10. L 111 > spatial variability of snow density. Please note what processes impact this ie is this snow compaction or from blowing snow?

Both compaction and transport of snow by the wind have been identified to have an effect on the heterogeneity of the snow cover. Details and references have been added in the manuscript: "the spatial variability of snow density that is expected over the sea ice floe due to the combined effects of wind-induced snow redistribution and compaction, especially in the vicinity of topographical features affecting the wind flow-field (Leonard and Maksym, 2011; Sturm and Massom, 2016)." [L140]

**3.5   Page 5**

1. Fig 1 > during a helicopter flight This is the first mention of a held. I thought this TLS was from the ship? Please clarify or remove.

Specifications about the ALS were indeed missing in the manuscript. We added details to make it more clear: "The topographical image in the background of Figure 1 is derived from aerial laser scan measurements taken on November 12, 2019 during an helicopter flight. It is only used to illustrate the relative sea ice floe location of the measurements used in our study. Throughout this manuscript the term "scans" only refer to the TLS observations." [L123]

2. L 120 > correction of orientation. I don't understand what this means. Please clarify what corrections were done.

We gave a more detailed summary of the data treatment achieved, but all of these should be precisely detailed and published in a data journal article linked to the final Met City data set over the next months. We'll add the reference as soon as we have it. Regarding the "correction of orientation", we meant: "Corrective rotations of the winds to be relative to true north" [L153]. This was added to the manuscript.

3. L 122 > among other instruments. Please list the source of all observations.

There were several instruments taking precipitation measurements during MOSAiC but we chose to use one of them in our simulations, the KAZR radar data as it has shown to be quite reliable. Therefore we consider that a complete description of the other instruments is not necessary. We changed in the text to: "At last, we used precipitation estimates recorded during MOSAiC using ship-based Ka-Band ARM (Atmospheric Radiation Measurement) Zenith Radar (KAZR) reflectivities" [L156]. It should be less confusing now.

4. L125 > that this data. This = Radar derived precip estimates in general or the data used here?

We are here talking about the specific data set measured during MOSAiC, which we specified in the manuscript: "the 280 m-ranged KAZR data measured during MOSAiC is on the whole reliable." [L161]

5. L127 > by releasing particles. Are all the particles same size? I think it is noted they have a distribution later but just note it here.

We added: "by releasing particles with log-normally distributed sizes in the domain" [L164].

**3.6 Page 6**

1. L 147 > walls. I know what you mean but I would like you to-clarify what you mean walls in a natural context to aid a non-domain modeller in reading this manuscript.

   We changed the term "walls" to "domain surfaces (internal or external)". [L184].

**3.7 Page 7**

1. L179 > which was added within the core code. By the authors? Ie is this part of the scientific contribution?

   Yes, it was added by the authors. We specified it in the manuscript as follows: "in the streamwise direction which was added by the authors within the core code of the solver and used as driver for the flow in the simulations presented hereafter" [L216].

**3.8 Page 8**

1. L188 > (Bagnold, 1941). Is this really the best reference? Bagnold 1941 is sand processes, not snow. There are more recent descriptions of the blowing snow transport outside of wind tunnels such as:

   *Aksamit, N. O. & Pomeroy, J. W. Near-surface snow particle dynamics from particle tracking velocimetry and turbulence measurements during alpine blowing snow storms. The Cryosphere 10, 3043–3062 (2016).*

   that should be referenced.

   We added references for snow transport and precised that Bagnold (1941) was about sand in Section 2.2.2. Snow-wind interaction model [L225]. It was indeed not clear in the original manuscript.

2. L195 > Bagnold (1941). Bagnold 1941 is sand, not snow. Odd to include it in this list. Would suggest replace it with the early efforts to adopt Bagnold to blowing snow that I list above.

   This is a good point. The whole paragraph was re-written with a clearer distinction between sand and snow transport. Additional references specific to snow are now present (e.g., Schmidt, 1986; Pomeroy and Gray, 1990; Li and Pomeroy, 1997) [L238].

3. L200 > a threshold value defined as (Bagnold, 1941). It is not clear to me why a sand-grain threshold is used. Where is the constant A from? Is this a snow value or a sand value?

   The A constant has first been defined by Bagnold (1941) for sand, but it has then been adapted to snow by Clifton et al. (2006) who tried to empirically define the best value for A (=0.18 after their study). It should be more clear now in the text: "Experimental results on snow transport initiation by Clifton et al. (2006) showed a good agreement with Bagnold's initial formulation of shear stress threshold, using A = 0.18 as an empirical constant" [L242].

4. L 207 > empirical parameter set to 1.5 (Doorschot and Lehning, 2002). I had assumed equation (11) was referring to eqn (8) in Doorschot and Lehning, (2002) but they note a value for $C_{ae}$=1. I don't see any other $C_{ae}$ in that manuscript so please double check this value.

   Good catch - this is a mistake. The value $C_{ae} = 1.5$ was set according to Groot Zwaaftink et al. 2014. We corrected it in the manuscript [L248].

**3.9 Page 9**

1. L 235 Note in this section the distribution and mean cell size of the hexes.

   We added the following cell volumes information in the text [L291]: Mean cell size: 0.126 $m^3$− Min: 0.019 $m^3$− Max: 0.277 $m^3$

**3.10 Page 10**

1. L 243 > with STereo Lithography (STL) input data. I know this is a file format but I think this should be clarified.

   A more detailed description of STL format was added to the text: "(...) input data in the STereo Lithography (STL) format. STL files describe the surface geometry of a three-dimensional object without any representation of other attributes (e.g. color, texture)" [L284].

2. L245 > the vertical grid spacing dz ranges between

   Are you using log spacing? Alternatively how were the vertical layer thicknesses decided?

   We did not use a log spacing but a linear stretching of the cell size from the surface to the top of the domain with a constant ratio of 1.12. The size of the cells was chosen based on the topography features needed to be captured and on the numerical stability of our simulations. We have made several trials with various mesh sizes. Cells that were too small were leading to numerical instabilities due to cell skewness or non-orthogonality in addition to enhanced computational time. Cells that were too big were not capturing well enough the topography. We chose a middle-ground between the two situations. It took a lot of trial and errors before finding a good set-up for our mesh. We had to compromise between precision and stable/computationally-affordable numerical simulations. We added details in the Section 2.3.1. Numerical Domain of the manuscript: "The cell size was made fine enough to accurately capture the topographical features found at the sea ice surface, while keeping a reasonable mesh size to perform the computations." [L289]

3. L 255 > corresponding cell of the connected periodic plane. Maybe note this results in snow being added on the upwind domain boundary.

   We agree. It is a simple but efficient way to explain things. We added the following explanation: "In other terms, this approach translates into snow being injected at the upwind domain boundary." [L299]

4. L 256 > at the wall. I would describe this for the reader who is interested in this approach but does not know what a wall means in this numerical context.

   The sentence was slightly changed to be more meaningful for non-CFD modellers: "Regarding the sea ice boundary at the bottom, no-slip and impermeability boundary conditions were imposed at its surface (...)" [L300]

5. L258 > set to a null value. Null = zero in this context? If so just say zero for clarity.

   Yes, we meant zero. The term was changed in the text [L303].

6. L 261 > for a neutral flow. Define what you mean by neutral flow – I assume neutral stability?

   Yes, we meant neutral stability. A reference was added in case the reader wants to know more on the approach: "for a flow with neutral stratification (Zhang, 2009)." [L306]

**3.11 Page 11**

1. L 273 > as much as possible.

   We removed the term [L330].

**3.12 Page 12**

1. L 290 > time of 1000s. 1000 s.

   Space was added after "1000" in the manuscript [L345].

2. L290 > particles aloft in the air got. "were" not got. The whole sentence was changed in the manuscript, as another method was used to derive the total duration snow distribution results [L344].

3. L290 > through the deactivation of the fluid driving force in snowBedFoam. This is not clear to me exactly what this mean. Please clarify what 'deactivation' means for a wind velocity. Are developing a particle steady-state and then turn the wind off and let it settle?

   Yes, what we meant is exactly what you have understood. However, we changed our strategy in the revised manuscript, explained above in this document (Section 2.4.3, using the steady-state rates). The results are not very different but it makes more sense theoretically. We changed it accordingly in the revised manuscript to "Based on the variations of the total snow mass aloft in the domain, the time step at which a steady-state is reached is identified (stable saltation flux) and the deposition and erosion rates in each cell are derived based on the snow mass distribution results for the steady-state period until the end of the simulation. The estimated snow distribution at the end of a given wind period are further obtained by multiplying the rates with the total duration of the simulated event." [L349]

4. L 302 > threshold formulation (Bagnold, 1941). Ah, so this is where A is defined. Please add this to the description of eqn (10).

   Yes, A is defined here. We agree that it is quite late in the text, thus we also added it when defining the shear stress threshold in Equation (10) [L243].

5. L 290 > the preferential deposition. Preferential deposition arises due to terrain impacts on local meteorological conditions, causing increased deposition on the leeward slopes and decreased deposition on the windward, and is typically a critical process in mountain terrain (e.g., Gerber et al., 2019; Mott et al., 2014; Vionnet et al., 2017). A few places in the manuscript it seems like blowing snow process and the deposition of suspended snow to be called 'preferential deposition'. I would like to see this tightened up so the reader is not potentially confused.

   The term was indeed wrongly used here. The whole sentence was changed in the manuscript making the term disappear, but we made sure that it was not confusing in the other parts of the manuscript.

6. L 300 > (280m range gate). 280 m, but also what is a range gate?

   Changed to "280 m" [L359]. We added further explanations for the range gate in section 2.1. MOSAiC campaign: "KAZR devices provide several intervals of range in the vertical (or time delay from transmission) within which returning radar signals are measured. Gating is used to isolate the echoes from different regions of distributed targets (Widener et al., 2012)." [L158]

**3.13   Page 14**

1. L327 > extrapolated areal. What does "extrapolated" mean in this context?

   We explain what we call "extrapolation method" at L286 of the manuscript (older version). By "extrapolated areal snow mass" we refer to the results obtained by multiplying the snow deposition rates by the duration of the wind events. We added a few words in the revised manuscript to clarify this, as follows: "The term "extrapolated" refers to the snow-mass distribution values obtained after multiplying the simulated snow transport rates by the total duration of each snow transport event." [L397]

2. L320 > Additional limitations arise from the forcing of the model. It is not totally clear to me not why simulate the whole time series? I realize 'compute' is offered up, but how prohibitively long would it be? It's probably outside the scope of this project (but perhaps not) however it would be interesting to know how much worse these assumptions made the model output v. running the model for the entirety of the observation period

   To give an idea of the computational effort needed, it takes about three weeks to simulate an event of 12 hours (i.e. 1.5 day for 3600 s of simulation, see Section 2.5 above). Therefore, simulating the whole week of observation would be extremely heavy computationally. The differences between results obtained with a fully-simulated event and the rate-derived method we employ were quantified above in Section 2.5.

3. L327 > Figures 4 and 5 report the results. Legend needs units (even if its in the caption).
   We added units in the Figures (pages 32, 33 of the revised manuscript).

**3.14   Page 15**

1. L336 > in blue in the surface friction velocity plots (Fig. 5). These low friction velocity streaks require more description and a quantification if the authors believe they are 'real'.

   We discussed this point in the Major Comments part of this document, section 2.3. A paragraph was added in the Results section of the revised manuscript, where we discuss these low-velocity streaks [L410] : "The surface friction velocity results in Figure 5 show elongated, low-velocity streaks for events I to IV. They are especially apparent in Figure 5-I due to the lower range of surface shear stress. (...)". The added text is a summary of what we explained above in Section 2.3.

**3.15   Page 16**

1. L 387 > may be multiple reasons for the. Density assumption is not addressed here.

   The density assumption is addressed later in the Discussion section, paragraph *1.Precipitation and other measurements* [L529].

2. L369 > Quantitatively, our model appears only partially successful. I would like to see RMSE + CV for the domain.

   The different grid scales between the measurements and the numerical results make it difficult to compute the RMSE, and it would also be dependent on the interpolation method employed. This is why we chose to represent the results with Figure 7 in the manuscript.

**3.16 Page 17**

1. L392 > many specific wind conditions. Would be good to muse on if the neglected conditions are similar to those shown by Aksamit 2016, for instance. I would also explicitly note "specific wind conditions" to include, e.g., gusts, etc.

   We included additional details on the mentioned "specific wind conditions" in the manuscript [L475] and a reference to the work of Aksamit and Pomeroy (2016) [L476].

2. L391 > used four averaged values for wind speed and direction in OpenFOAM to represent a one-week period of measurements. I am not I surprised this didn't when we know blowing snow is at higher temporal scale!

   Yes, this point was discussed in more details in Section 2.5 of the present document. We explain in more details the motivation of our approach in the manuscript [L478, 483].

L 420 > Could some of this be due to ignoring sublimation? Figure 6 (left) suggests an over estimation of deposition in areas such as due north of the middle of B, on the flatter (?) section. The elevation (?) isolines between these two figures are different though, making a qualitative comparison difficult.

It could be, but it is difficult to verify the assumption due to the large amount of factors impacting the snow mass balance. To still mention the hypothesis, we discuss the sublimation processes in Section 4 of the revised manuscript [L542]: "Moreover, thermal processes such as sublimation were considered to be negligible and this may have impacted the snow mass deposition in the measurements, by lowering the overall snow deposition in flatter areas."

**3.17 Page 18**

1. L425 > tendency to overestimate precipitation. By how much, exactly?

   By about 10 % (Wagner et al., 2021) during the MOSAiC campaign. However, it is difficult to precisely quantify this value, which only constitutes a rough estimate. It has been shown that the uncertainty related to the KAZR-derived snowfall rate can reach up to 50% (Matrosov, Shupe, and Uttal 2022). If we compare the KAZR measurements for the storm event of interest in this study, we find that the measured rates are twice as high as the device recording the lowest precipitation rates for the same period. But the "device recording the lowest precipitation rates" (and all the other devices mentioned in Section 2.4. of this document) also have their own biases. Therefore, giving an exact value for overestimation remains difficult. This point is mentioned in the revised manuscript [L516].

**3.18 Page 20**

1. L496 > snow distribution patterns were accurately captured. You note the following in the. results section "we observe that the quantitative performance of our model is not optimal" and Fig 6 shows substantial differences.

   We've tried to make a distinction between *qualitative* and *quantitative* aspects throughout our manuscript (also in the Discussion). Thus, when we mention "we observe that the quantitative performance of our model is not optimal" we express that the results are not satisfying from a quantitative point of view. On the other hand, the location of the snow deposition and erosion patches have been accurately reproduced by the model (in the lee of the ridges). This is what we wanted to highlight by using the term "patterns": we are here talking about the

qualitative aspects exclusively, shown by the use of "Qualitatively, (...)" at the beginning of the same sentence [L598].

2. L498 > enhanced deposition. Blowing snow deposition, correct?

The term "deposition" encompasses here the preferential deposition of precipitation, as Event IV simulated a precipitation event, and the deposition of blowing snow. We moved the term "enhanced" here because we realized it was not at the right location in the text [L601].

3. L 505 > performance. Word choice – ensure it is clear you mean the accuracy of the model output and not the computational performance of the model.

We added the term "quantitative" before performance to prevent any confusion by the reader [L608].

**3.19    Page 26**

1. I would like to see units on the colorbars.

It would indeed increase the readability of the plots. We added the units in the revised manuscript, as well as a legend on top of the color bar.

**3.20    Page 28**

1. extrapolated snowBedFoam 1.0. I don't understand what extrapolated means for these results.

We explained better what we meant by "extrapolated" in the Results section [L397]. It refers to the values obtained by multiplying the simulated snow deposition/erosion rates with the respective duration of the events. Hopefully it is clear now by reading the revised text.

**References**

Sørensen, Michael (2004). "On the rate of aeolian sand transport". In: *Geomorphology* 59.1. Aeolian Research: processes, instrumentation, landforms and palaeoenvironments, pp. 53–62. ISSN: 0169-555X. DOI: https://doi.org/10.1016/j.geomorph.2003.09.005. URL: https://www.sciencedirect.com/science/article/pii/S0169555X03003131.

Clifton, Andy, Jean-Daniel Rüedi, and Michael Lehning (Dec. 2006). "Snow saltation threshold measurements in a drifting-snow wind tunnel". In: *Journal of Glaciology* 52, pp. 585–596. DOI: 10.3189/172756506781828430.

Mott, Rebecca and Michael Lehning (Aug. 2010). "Meteorological Modeling of Very High-Resolution Wind Fields and Snow Deposition for Mountains". In: *Journal of Hydrometeorology - J HY-DROMETEOROL* 11, pp. 934–949. DOI: 10.1175/2010JHM1216.1.

Chung, Yi-Ching, Stéphane Bélair, and Jocelyn Mailhot (2011). "Blowing Snow on Arctic Sea Ice: Results from an Improved Sea Ice–Snow–Blowing Snow Coupled System". In: *Journal of Hydrometeorology* 12.4, pp. 678–689. DOI: 10.1175/2011JHM1293.1. URL: https://journals.ametsoc.org/view/journals/hydr/12/4/2011jhm1293_1.xml.

Leonard, Katherine C. and Ted Maksym (2011). "The importance of wind-blown snow redistribution to snow accumulation on Bellingshausen Sea ice". In: *Annals of Glaciology* 52.57, pp. 271–278. DOI: 10.3189/172756411795931651.

Wong, Kai Chi (2012). "Performance of Several Present Weather Sensors as Precipitation Gauges". In.

Groot Zwaaftink, Christine et al. (Oct. 2014). "Modelling Small-Scale Drifting Snow with a La-
grangian Stochastic Model Based on Large-Eddy Simulations". In: *Boundary-Layer Meteorology*
153. DOI: 10.1007/s10546-014-9934-2.

Trujillo, Ernesto et al. (Oct. 2016). "Changes in Snow Distribution and Surface Topography Fol-
lowing a Snowstorm on Antarctic Sea Ice". In: *Journal of Geophysical Research: Earth Surface*
121. DOI: 10.1002/2016JF003893.

Hesp, Patrick A. and Thomas A. G. Smyth (2017). "Nebkha flow dynamics and shadow dune
formation". In: *Geomorphology* 282, pp. 27–38.

Ivanell, S. et al. (2018). "Micro-scale model comparison (benchmark) at the moderately complex
forested site Ryningsnäs". In: *Wind Energy Science* 3.2, pp. 929–946. DOI: 10.5194/wes-3-
929-2018. URL: https://wes.copernicus.org/articles/3/929/2018/.

Mott, Rebecca, Vincent Vionnet, and Thomas Grünewald (2018). "The Seasonal Snow Cover Dy-
namics: Review on Wind-Driven Coupling Processes". In: *Frontiers in Earth Science* 6, p. 197.
ISSN: 2296-6463. DOI: 10.3389/feart.2018.00197. URL: https://www.frontiersin.org/
article/10.3389/feart.2018.00197.

Sharma, V., F. Comola, and M. Lehning (2018). "On the suitability of the Thorpe–Mason model
for calculating sublimation of saltating snow". In: *The Cryosphere* 12.11, pp. 3499–3509. DOI:
10.5194/tc-12-3499-2018. URL: https://www.the-cryosphere.net/12/3499/2018/.

Comola, F. et al. (2019). "Preferential Deposition of Snow and Dust Over Hills: Governing Processes
and Relevant Scales". In: *Journal of Geophysical Research: Atmospheres* 124.14, pp. 7951–7974.
DOI: 10.1029/2018JD029614.

Wagenbrenner, Natalie S. et al. (2019). "Development and Evaluation of a Reynolds-Averaged
Navier–Stokes Solver in WindNinja for Operational Wildland Fire Applications". In: *Atmo-
sphere* 10.11. ISSN: 2073-4433. DOI: 10.3390/atmos10110672. URL: https://www.mdpi.com/
2073-4433/10/11/672.

Hersbach, Hans et al. (2020). "The ERA5 global reanalysis". In: *Quarterly Journal of the Royal
Meteorological Society* 146.730, pp. 1999–2049. DOI: https://doi.org/10.1002/qj.3803.
eprint: https://rmets.onlinelibrary.wiley.com/doi/pdf/10.1002/qj.3803. URL:
https://rmets.onlinelibrary.wiley.com/doi/abs/10.1002/qj.3803.

Wagner, D. N. et al. (2021). "Snowfall and snow accumulation processes during the MOSAiC winter
and spring season". In: *The Cryosphere Discussions* 2021, pp. 1–48. DOI: 10.5194/tc-2021-
126. URL: https://tc.copernicus.org/preprints/tc-2021-126/.

Webster, M. A. et al. (2021). "Snow on Arctic Sea Ice in a Warming Climate as Simulated in
CESM". In: *Journal of Geophysical Research: Oceans* 126.1. e2020JC016308 2020JC016308,
e2020JC016308. DOI: https://doi.org/10.1029/2020JC016308. eprint: https://agupubs.
onlinelibrary.wiley.com/doi/pdf/10.1029/2020JC016308. URL: https://agupubs.
onlinelibrary.wiley.com/doi/abs/10.1029/2020JC016308.

Matrosov, S.Y., M. D. Shupe, and T. Uttal (2022). "High temporal resolution estimates of Arctic
snowfall rates emphasizing gauge and radar-based retrievals from the MOSAiC expedition". In:
*Elementa: Science of the Anthropocene* 10.1. DOI: 10.1525/elementa.2021.00101.

Melo, D. B. et al. (2022). "Modeling Snow Saltation: The Effect of Grain Size and Interparticle Cohe-
sion". In: *Journal of Geophysical Research: Atmospheres* 127.1. e2021JD035260 2021JD035260,
e2021JD035260. DOI: https://doi.org/10.1029/2021JD035260. eprint: https://agupubs.
onlinelibrary.wiley.com/doi/pdf/10.1029/2021JD035260. URL: https://agupubs.
onlinelibrary.wiley.com/doi/abs/10.1029/2021JD035260.

**Responses to Referee 2 for the manuscript:**

**"Modelling the small-scale deposition of snow onto structured Arctic sea ice during a MOSAiC storm using snowBedFoam 1.0."**

April 7, 2022

**1  General answer**

The present document provides an answer to the second referee, who posted his/her comments on January 3, 2022 in the open discussion of our GMD manuscript. We thank the reviewer for his/her thoughtful comments and efforts towards improving our manuscript as well as for the expertise provided. The references to the lines in the revised manuscript are included within square brackets, such as [L00]. In the author's track-changes file, the changes based on the comments of the 1st referee (posted on October 5th, 2021) are highlighted in blue color. The changes in green color are related to the remarks of the second referee (posted on January 3, 2022) and the violet color are the changes related to the comments of all referees. The additional changes added by the authors appear in gray. Note that some values for the forcing parameters have been changed, based on the most recently processed meteorological data (Table 1 of the revised manuscript). In the present document, the remarks of the referee are in black color and the author comments are in green.

**2  Major comments**

For the sake of clarity, we chose to divide the major comments in sub-sections and named them after their content.

**2.1  Wind velocity modelling at the boundary**

"The finest vertical grid is 0.1m, while the saltation layer of blowing snow is almost the same order of magnitude. It means almost all the interactions between snow particles and airflow are in the first of the vertical grid, which indicates that the wind velocity estimation on particle location is important. Could the authors add more detail about the wind velocity approach where the snow particles located."

This is, indeed, an important point. The wind velocity in the near-wall region is modelled through a wall function from the standard OpenFOAM library. Wall functions satisfy the physics near the wall and aim to bridge the inner region between the wall and the fully developed turbulent region. We made use of the so-called *epsilonLowReWallFunction* and *kLowReWallFunction* wall functions in OpenFOAM, which are adapted to both low and high Reynolds numbers and operate

in two modes based on the cell size, viscosity and friction velocity ratio (y+). With this approach, the boundary layer does not need to be resolved, which significantly reduces the computational domain and mesh size. Wall functions have been commonly used in other snow transport models (e.g. Kang et al. 2018; V. Sharma, Comola, and M. Lehning 2018). More details regarding the OpenFOAM wall functions are available in Fangqing 2016.

In our modelling framework, any Eulerian quantity needed for the Lagrangian calculations (e.g. flow velocity) is interpolated based on the particle location (Smith and Ebert 2010; Tofighian, Amani, and Saffar-Avval 2019). We used the so-called *"cellPoint"* interpolation method in Open-FOAM, which works by dividing the cells into tetrahedrons connecting the cell center and the faces, then determining which tetrahedron encloses the parcel location, and finally performing linear interpolation using inverse distance weights (Leonard, Qiao, and Nabi 2021). We added more information about the wall functions and interpolation method to the parcel location in Section 2.3.3. Numerics of the manuscript [L301, L321].

**2.2 Eulerian and Lagrangian timesteps**

"What are the time steps for airflow and particle, respectively? Since this model considers the splash process, the time step for the particle should be very limited to a small value."

For the flow, we make use of an automatic time step control called *"adjustableRunTime"* available in OpenFOAM, which adapts the time step based on a maximum Courant number value defined by the user (in our case, it was set to 1). The Courant number as the stability criterion is defined as

$$Co = \frac{\mathbf{U}^f \Delta t_f}{L} \tag{1}$$

where $\mathbf{U}^f$ is the fluid velocity, $L$ is the numerical cell length scale defined as the ratio between the cell volume and the cell surface area, and $\Delta t_f$ is the fluid-phase time step. This definition implies that the Eulerian time step changes with the grid size and local velocity. More information regarding the adjustable time step method for the flow is available in Moukalled, Mangani, and Darwish 2015 and Jafari, Varun Sharma, and Michael Lehning 2022.

There are various factors impacting the parcel (Lagrangian) time step. Within one Eulerian time step ($\Delta t_f$), there are several smaller Lagrangian time steps ($\Delta t_p$), which allow to adequately capture the parcel motion. In our Eulerian-Lagrangian model, the so-called "face-to-face tracking algorithm" (Peng 2008; Macpherson, Nordin, and Weller 2009) ensures that a parcel cannot cross a cell boundary without updating its properties accordingly. In other words, as soon as a parcel encounters a face including cell boundary faces, functions related to the face type are called within the solver. If a parcel reaches the surface (sea ice) boundary, the rebound-splash function is automatically called (Equations 12 to 14 in the manuscript) and the splashing process is fully resolved. Thus, the face-to-face algorithm adapts the Lagrangian time step depending on the crossed boundaries and automatically limits its value to appropriately capture micro-scale processes such as the rebound-splash of snow grains at the surface. We should mention that the motion of parcels is also controlled by a maximum Courant number specific for parcels, which is defined similarly to Equation 1 but with replacing the fluid velocity and timestep by the equivalent parcel quantities ($\mathbf{U}^p$ and $\Delta t_p$). In OpenFOAM, the maximum Courant number value for parcels is set by default to 0.3. Details on the computation of the Eulerian and Lagrangian time steps were added to Section 2.3.3. Numerics of the manuscript [L312].

**2.3   Simulation period**

"The period of time is a week, which is really long for blowing snow evaluation."

We chose our one-week simulation period based on the available terrestrial laser scans measured during the MOSAiC expedition. To be able to compare the model results to snow distribution data obtained in the field, we had no choice but to simulate a full week of snow redistribution on sea ice. To limit the computational effort to a reasonable amount, we selected the dominating wind phases within that week using a friction velocity threshold (0.2 m/s) and came up with the method of computing the erosion and deposition rates during the first seconds of simulation and then using the latter to estimate the total snow distribution after several hours. It is based on the assumption that a flow-particle equilibrium state (steady-state) is reached in the simulations, which we verified by analyzing the low fluctuations of the total particle mass in the system. Figure 1 shows an example of the total mass in the system for the first event described in the manuscript (November 6 2019, from 12:00 to 23:00). We are aware that this method has strong limitations but it seems like the most appropriate choice to simulate such a long period, given the modelling framework employed in our study. We added the following "Although long for the evaluation of blowing snow, we defined our simulation period based on the terrestrial laser scans measurements available during the MOSAiC expedition. To be able to compare the model results to snow distribution data obtained in the field, we had no choice but to simulate a full week of snow redistribution on sea ice." in Section 2.4. Modelling assumptions of the manuscript [L387].

[Figure]

Figure 1: Total snow mass in the system (kg) over simulation time for event I.

**2.4 Thermal processes**

"The authors may also need to discuss the effects of other snow processes on snow distribution, such as the thermal processes."

We replied to the concerns of the first referee about the consideration of sublimation processes (blowing snow and surface sublimation) in our model. The detailed reply can be read in Section 2.1. Snow Sublimation of the document "*Responses to Referee 1 for the manuscript*" available within the author's response document. We refer to this section which also provides an answer to the comments of the second referee regarding the consideration of thermal processes. In summary, given the location and time of interest of our simulations, we consider the sublimation fluxes to be negligible. Additional comments were added to the revised version of the manuscript, such as: "Only pure mechanical fluid-particle interactions are considered here, thus we distinctively evaluate the impact of the horizontal snow transport on the sea ice snow mass balance at a given location. Thermal processes such as the sublimation of blowing snow and snow at the surface, although having a big role in the snow mass budget at certain spatio-temporal scales, are assumed to be negligible given the time period and location of interest (Chung et al. (2011); Webster et al. (2021))" [L91]. Furthermore, a detailed explanation of our hypothesis is given in Section 2.4. Modelling assumptions of the revised manuscript [L373]. The topic is also further discussed in Section 4 [L542].

**References**

Peng, Fabian Kärrholm (2008). "Numerical Modelling of Diesel Spray Injection, Turbulence Interaction and Combustion". In.

Macpherson, Graham B., Niklas Nordin, and Henry G. Weller (2009). "Particle tracking in unstructured, arbitrary polyhedral meshes for use in CFD and molecular dynamics". In: *Communications in Numerical Methods in Engineering* 25.3, pp. 263–273. DOI: https://doi.org/10.1002/cnm.1128. eprint: https://onlinelibrary.wiley.com/doi/pdf/10.1002/cnm.1128. URL: https://onlinelibrary.wiley.com/doi/abs/10.1002/cnm.1128.

Smith, William G and Michael P Ebert (2010). *A method for unstructured mesh-to-mesh interpolation*. Tech. rep. Naval Surface Warfare Center Carderock Div Bethesda MD.

Moukalled, F., L. Mangani, and M. Darwish (2015). *The Finite Volume Method in Computational Fluid Dynamics: An Advanced Introduction with OpenFOAM and Matlab*. 1st. Springer Publishing Company, Incorporated. ISBN: 3319168738.

Fangqing, Liu (2016). "A Thorough Description Of How Wall Functions Are Implemented In OpenFOAM". In: *Proceedings of CFD with OpenSource Software*. URL: http://www.tfd.chalmers.se/~hani/kurser/OS_CFD_2016.

Kang, Luyang et al. (2018). "CFD simulation of snow transport over flat, uniformly rough, open terrain: Impact of physical and computational parameters". In: *Journal of Wind Engineering and Industrial Aerodynamics* 177, pp. 213–226. ISSN: 0167-6105. DOI: https://doi.org/10.1016/j.jweia.2018.04.014.

Sharma, V., F. Comola, and M. Lehning (2018). "On the suitability of the Thorpe–Mason model for calculating sublimation of saltating snow". In: *The Cryosphere* 12.11, pp. 3499–3509. DOI: 10.5194/tc-12-3499-2018. URL: https://www.the-cryosphere.net/12/3499/2018/.

Tofighian, H., E. Amani, and M. Saffar-Avval (2019). "Parcel-number-density control algorithms for the efficient simulation of particle-laden two-phase flows". In: *Journal of Computational Physics*

387, pp. 569–588. ISSN: 0021-9991. DOI: https://doi.org/10.1016/j.jcp.2019.02.052. URL: https://www.sciencedirect.com/science/article/pii/S002199911930169X.

Leonard, Eric, Hongtao Qiao, and Saleh Nabi (2021). "A Comparison of Interpolation Methods in Fast Fluid Dynamics". In: *International High Performance Buildings Conference* Paper 341. ISSN: 0167-6105. DOI: https://docs.lib.purdue.edu/ihpbc/341.

Jafari, Mahdi, Varun Sharma, and Michael Lehning (2022). "Convection of water vapour in snowpacks". In: *Journal of Fluid Mechanics* 934, A38. DOI: 10.1017/jfm.2021.1146.